# A FRET sensor of C-terminal movement reveals VRAC activation by plasma membrane DAG signaling rather than ionic strength

Benjamin König[1], Yuchen Hao[2,3,4], Sophia Schwartz[1], Andrew JR Plested[2,3,4], Tobias Stauber[1]*

[1]Institute of Chemistry and Biochemistry, Freie Universität Berlin, Berlin, Germany; [2]Institute of Biology, Humboldt Universität zu Berlin, Berlin, Germany; [3]Leibniz Forschungsinstitut für Molekulare Pharmakologie (FMP), Berlin, Germany; [4]NeuroCure, Charité Universitätsmedizin, Berlin, Germany

**Abstract** Volume-regulated anion channels (VRACs) are central to cell volume regulation. Recently identified as hetero-hexamers formed by LRRC8 proteins, their activation mechanism remains elusive. Here, we measured Förster resonance energy transfer (FRET) between fluorescent proteins fused to the C-termini of LRRC8 subunits. Inter-subunit FRET from LRRC8 complexes tracked VRAC activation. With patch-clamp fluorometry, we confirmed that the cytoplasmic domains rearrange during VRAC opening. With these FRET reporters, we determined VRAC activation, non-invasively, in live cells and their subcompartments. Reduced intracellular ionic strength did not directly activate VRACs, and VRACs were not activated on endomembranes. Instead, pharmacological manipulation of diacylglycerol (DAG), and protein kinase D (PKD) activity, activated or inhibited plasma membrane-localized VRACs. Finally, we resolved previous contradictory reports concerning VRAC activation, using FRET to detect robust activation by PMA that was absent during whole-cell patch clamp. Overall, non-invasive VRAC measurement by FRET is an essential tool for unraveling its activation mechanism.
DOI: https://doi.org/10.7554/eLife.45421.001

*For correspondence:
tobias.stauber@fu-berlin.de

**Competing interests:** The authors declare that no competing interests exist.

## Introduction

Volume regulation is essential for virtually all cell types, not only to counteract osmotic swelling and shrinkage but also during various processes like proliferation, migration and apoptosis (*Hoffmann et al., 2009*). Volume-regulated anion channels (VRACs) are ubiquitously expressed and involved in regulatory volume decrease (RVD) of vertebrate cells (*Chen et al., 2019*; *Jentsch, 2016*; *Stauber, 2015*). These channels open slowly following osmotic cell swelling and mediate the extrusion of chloride and organic osmolytes, leading to the efflux of water and hence cell shrinkage.

VRACs are composed of LRRC8 heteromers containing the essential subunit LRRC8A and at least one other LRRC8 protein, LRRC8B-E (*Qiu et al., 2014*; *Voss et al., 2014*). The subunit composition determines various biophysical properties and substrate permeability (*Planells-Cases et al., 2015*; *Syeda et al., 2016*; *Voss et al., 2014*). Although unlikely to be of physiological relevance, LRRC8A homomers can form anion channels, but with weak volume-sensitivity and very low conductance (*Deneka et al., 2018*; *Syeda et al., 2016*; *Yamada and Strange, 2018*). LRRC8 proteins form hexamers with the N-terminal halves of the six LRRC8 proteins constituting the transmembrane domain of the channel complex with a single pore. The C-terminal halves contain at least 16 leucine-rich repeats that form horseshoe-like structures reaching into the cytoplasm and their apparent flexibility

has been hypothesized to be related to channel gating (*Deneka et al., 2018*; *Kasuya et al., 2018*; *Kefauver et al., 2018*; *Kern et al., 2019*).

Prior to cloning, the biophysical properties of VRAC-mediated chloride currents (I$_{Cl,swell}$) and osmolyte conductance were studied extensively using mainly electrophysiological and pharmacological methods (*König and Stauber, 2019*; *Pedersen et al., 2016*; *Strange et al., 2019*); but their activation mechanism remained enigmatic for decades, in the absence of an unequivocal molecular identity for the channel. Numerous studies proposed contributions from membrane tension, the actin cytoskeleton and signaling pathways involving reactive oxygen species, calcium, G-proteins and phosphorylation cascades (*Akita and Okada, 2014*; *Chen et al., 2019*; *Pedersen et al., 2016*). Notably, the reduction of intracellular ionic strength (Γ$_i$) was reported to be crucial for VRAC activation upon osmotic cell swelling (*Voets et al., 1999*) and reconstituted LRRC8 complexes in lipid bilayers could indeed be activated by low ionic strength (*Syeda et al., 2016*). However, VRACs can also be activated under iso-osmotic conditions that are unlikely to encompass any changes in Γ$_i$ (*Pedersen et al., 2016*; *Strange et al., 2019*).

Here, we use intra-complex Förster resonance energy transfer (FRET) to monitor VRAC activity in a spatio-temporal manner. This method demonstrates movement of the LRRC8 C-termini and provides insights into the activation mechanism of VRACs.

## Results

### Inter-subunit FRET shows C-terminal movement during VRAC activation

Following the hypothesis of a conformational rearrangement of the LRRC8 C-terminal domains during VRAC activation, we aimed to monitor VRAC activity by inter-subunit FRET (*Figure 1A*). We first used acceptor photo-bleaching to test for FRET between cyan- and yellow-fluorescent proteins (CFP/YFP) fused to the C-termini of VRAC subunits. In HeLa cells co-expressing CFP- and YFP tagged LRRC8A (A-CFP/A-YFP), bleaching of YFP (the FRET acceptor) robustly increased the intensity of CFP (the FRET donor), compared to unbleached cells in the same field of view (*Figure 1B,C*). The same was observed when LRRC8A was combined with the VRAC subunit LRRC8E (A-CFP/E-YFP or A-YFP/E-CFP), but not when combined with plasma membrane-localized CD4-YFP, used as control for FRET by membrane crowding (*Figure 1C*). Therefore, irrespective of VRAC subunit composition, the C-termini are within FRET distance for the fluorescent protein adjuncts (~5 nm).

Next, we explored whether FRET changes during VRAC activation. We used sensitized-emission FRET and monitored the corrected FRET (cFRET, see Materials and methods) values in HeLa cells expressing A-CFP/E-YFP (*Figure 1D*). Switching from isotonic (340 mOsm) to hypotonic (250 mOsm) buffer led to a robust decrease in cFRET by 5–10% within 90 s in both HeLa and HEK293 cells (*Figure 1E*). In contrast, the intra-molecular cFRET of a cytosolic tandem FRET pair construct, CFP-18aa-YFP (*Elder et al., 2009*), and of a plasma membrane-localized ionotropic glutamate receptor (*Zachariassen et al., 2016*) were unaffected by hypotonic buffer (*Figure 1E*, *Figure 1—figure supplement 1*). These observations rule out FRET artifacts from changes of osmolarity or ion concentrations. To better understand the cFRET changes with the activation of VRAC, we expressed fluorescently-labelled A-CFP/E-YFP (in this case Cerulean/Venus, see Materials and methods) in HEK293 cells deficient for all five LRRC8 genes (HEK293 KO) (*Lutter et al., 2017*) and monitored cFRET and whole-cell currents simultaneously using patch-clamp fluorometry (*Zheng and Zagotta, 2003*). The kinetics of the cFRET drop and its reversibility correlated with the activation and inactivation of hypotonicity-induced VRAC-mediated anion currents (*Figure 1F*), demonstrating that the cFRET decrease indeed reflects VRAC opening. The current was in addition to any potential residual activity at isotonicity due to the fluorescent tags (*Gaitán-Peñas et al., 2016*). A progressive decrease in cFRET accompanied additional hypotonicity but, again mirroring VRAC, cFRET did not increase with hypertonic treatment (*Figure 1G*, *Figure 1—figure supplement 2A*). We frequently observed varying extents of intracellular retention of LRRC8A or LRRC8E. We did not see a correlation between intensity of CFP or YFP and absolute cFRET values (not shown), neither did relative changes in cFRET depend on the A-CFP/E-YFP expression ratio (*Figure 1—figure supplement 2B*). Remarkably, cFRET also decreased for A-CFP/A-YFP constructs (*Figure 1E*, *Figure 1—figure supplement 1*), providing evidence for activation of homomeric LRRC8A complexes in a cellular context.

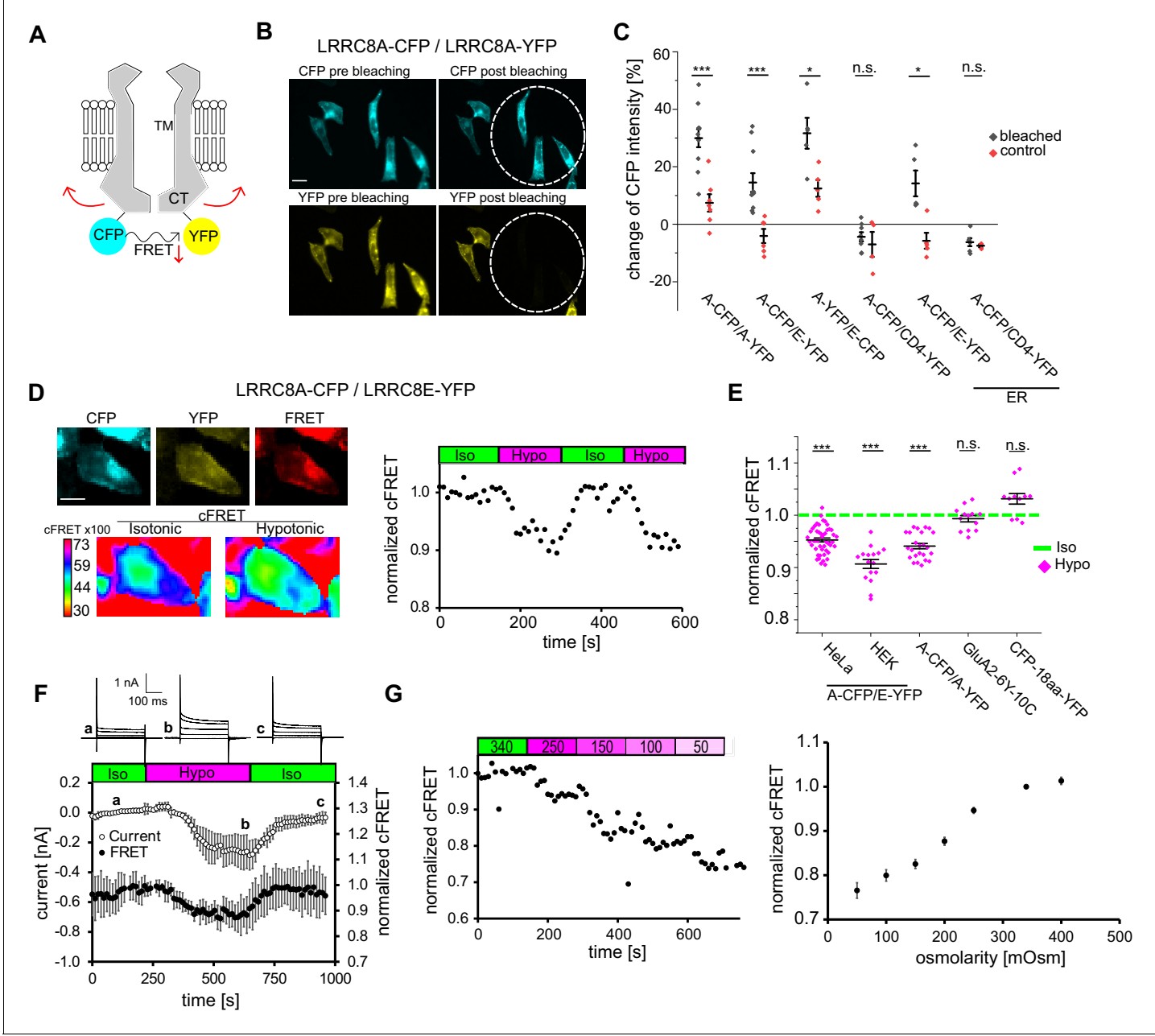

**Figure 1.** FRET between fluorescent proteins fused to LRRC8 C-termini and its changes with VRAC activity. (**A**) Schematic view of two VRAC subunits of a hexamer fused with the FRET pair CFP and YFP at their C-termini. Hypothetical movement of the intracellular C-terminal domain leads to FRET changes. (**B**) Wide-field images of HeLa cells expressing A-CFP/A-YFP before and after bleaching YFP. White circles indicate the size of the closed field diaphragm to bleach this part of the FOV. Scale bar, 20 μm. (**C**) Quantification of acceptor photo-bleach experiments. For each experiment, CFP intensity was measured before and after bleaching for individual cells within (bleached, black diamonds) and outside (control, red diamonds) of the bleached part of the FOV. ER: overexpressed proteins were trapped in the endoplasmic reticulum by 5 μg/ml BFA during expression. Data represent percentage of CFP intensity change for individual cells (diamonds) and mean ± s.e.m. (**D**) Representation of an seFRET experiment with an A-CFP/E-YFP-expressing cell. Left: images of the acquired channels (donor CFP, acceptor YFP, FRET) with the calculated cFRET map for two time points in isotonic and hypotonic buffer below. Scale bar, 10 μm. Images were acquired in 10 s intervals with 8 × 8 binning. Right: normalized cFRET values during buffer exchange experiment. (**E**) Measurements of cFRET, normalized to the value in isotonic buffer, when switching to hypotonic buffer for cells expressing A-CFP/E-YFP (n = 9 dishes with 47 HeLa cells; n = 8 with 29 HEK293 cells), A-CFP/A-YFP (n = 8, 24 HeLa cells), glutamate receptor GluA2-6Y-10C (n = 4, 13 HEK293 cells) and CFP-18aa-YFP (n = 4, 11 HeLa cells). Data represent individual cells (diamonds) and mean ± s.e.m. (**F**) Simultaneous measurements of whole-cell current at −80 mV (open circles) and normalized cFRET values (solid circles) during buffer exchange experiment with A-CFP/E-YFP-expressing HEK293 KO cells. Data represent mean of 4 cells ± s.d. Above, representative current traces from voltage-step protocols at the indicated time points (a, b, c). (**G**) Left: normalized cFRET in a time-course for a single cell with application of decreasing osmolarities (as indicated

*Figure 1 continued on next page*

*Figure 1 continued*

above in mOsm). Right: summary of experiments with different osmolarity steps to cover the titration curve from 50 to 400 mOsm. In total, 68 cells were measured for the titration curve, n = 7–21 depending on osmolarity. Data represent mean ± s.e.m. Statistics: *p<0.05; ***p<0.0005; n.s., not significant, Student's *t*-test, comparing bleached and control (C) or hypotonic with isotonic (E).

DOI: https://doi.org/10.7554/eLife.45421.002

The following source data and figure supplements are available for figure 1:

**Source data 1.** Statistics of acceptor-bleaching experiment and hypotonicity-induced FRET changes.

DOI: https://doi.org/10.7554/eLife.45421.005

**Figure supplement 1.** Representative seFRET measurements during buffer exchange experiments with individual HeLa cells expressing A-CFP/A-YFP or CFP-18aa-YFP and a HEK293 cell expressing GluA2-6Y-10C (out of 24, 13 and 11 cells, respectively, for quantification in *Figure 1E*). cFRET values for CFP-18aa-YFP and GluA2-6Y-10C are unaffected by hypotonic treatment.

DOI: https://doi.org/10.7554/eLife.45421.003

**Figure supplement 2.** Hypertonicity does not affect FRET; hypotonicity-induced cFRET changes are independent of the A-CFP/E-YFP expression ratio.

DOI: https://doi.org/10.7554/eLife.45421.004

Overall, these cFRET measurements show that the C-termini of LRRC8 proteins in VRAC complexes rearrange during activation and that this can be exploited to monitor their activity in living cells via FRET microscopy.

## Reduced ionic strength does not activate VRAC on endomembranes

Reduced ionic strength is sufficient to activate reconstituted VRACs in lipid bilayer droplets (*Syeda et al., 2016*). To measure intracellular ionic strength, we used a FRET-based sensor that reports the electrostatic attraction between arginine and aspartate residues in opposed α-helices ('RD' sensor) (*Liu et al., 2017*). As expected, this sensor indicated a reversible decrease in $\Gamma_i$ in HeLa cells during hypotonic treatment (*Figure 2A*). Notably, the decrease was homogeneous over the entire cell volume (*Figure 2B*, *Figure 2—video 1*). If a decline in $\Gamma_i$ were sufficient to activate VRAC, then we hypothesized that it should already do so in the secretory pathway. To test for this, we trapped FRET-endowed VRACs in the endoplasmic reticulum (ER) by treatment with brefeldin-A (BFA) during expression (*Figure 2C*). FRET and hence complex formation in the ER of A-CFP/E-YFP (but not A-CFP/CD4-YFP) was confirmed by acceptor bleaching (*Figure 1C*). However, ER-trapped VRACs surprisingly did not respond to hypotonic treatment (*Figure 2D*). This was likely not due to incomplete glycosylation since mutation of the LRRC8A N-glycosylation sites (*Voss et al., 2014*) did not impinge on the activation-dependent cFRET drop at the plasma membrane (*Figure 2D*).

Intrigued by this finding, we set out to follow VRACs along the secretory pathway. To this end, we exploited a reverse aggregation system and fused two self-dimerizing mutated FK506-binding protein (FM) domains (*Rollins et al., 2000*) to the C-terminus of A-CFP. Inter-molecular dimerization of FM domains should lead to aggregation of VRACs in the ER, and upon addition of D/D solubilizer they should disaggregate and travel in bulk along the secretory route to the plasma membrane (*Figure 3A*). Indeed, initially aggregated A-CFP-FM2 expressed alone (*Figure 3B*, *Figure 3—video 1*) or with E-YFP (not shown) spread homogenously over the ER within 10 min in presence of D/D solubilizer, peaked in the Golgi apparatus at 80 min and displayed clear plasma membrane localization after 165 min. Concurrent sensitized FRET measurements showed that neither ER-localized (at 10 min) nor Golgi-localized (80 min) A-CFP-FM2/E-YFP VRAC responded to application of hypotonic buffer. On the other hand, as expected, cFRET was reduced for A-CFP-FM2/E-YFP that had reached the plasma membrane after 165 min (*Figure 3C*). Thus, despite hypotonic treatment reducing $\Gamma_i$ homogenously over the whole cell, VRACs are only activated when they reach the plasma membrane.

## Robust VRAC activation by PMA is blocked by whole-cell patch clamp

Next, we asked which plasma membrane-specific factors might be required for VRAC activation. First, we tested the influence of the actin cytoskeleton and cholesterol content, both of which have previously been implicated in cell volume regulation (*Byfield et al., 2006*; *Klausen et al., 2006*). In agreement with earlier reports on $I_{Cl,swell}$ in lymphocytes (*Levitan et al., 1995*), depolymerization of the actin cytoskeleton by latrunculin B did not alter VRAC activation by ~25% hypotonicity (*Figure 3—figure supplement 2A,B*). In contrast, cholesterol depletion by methyl-β-cyclodextrin

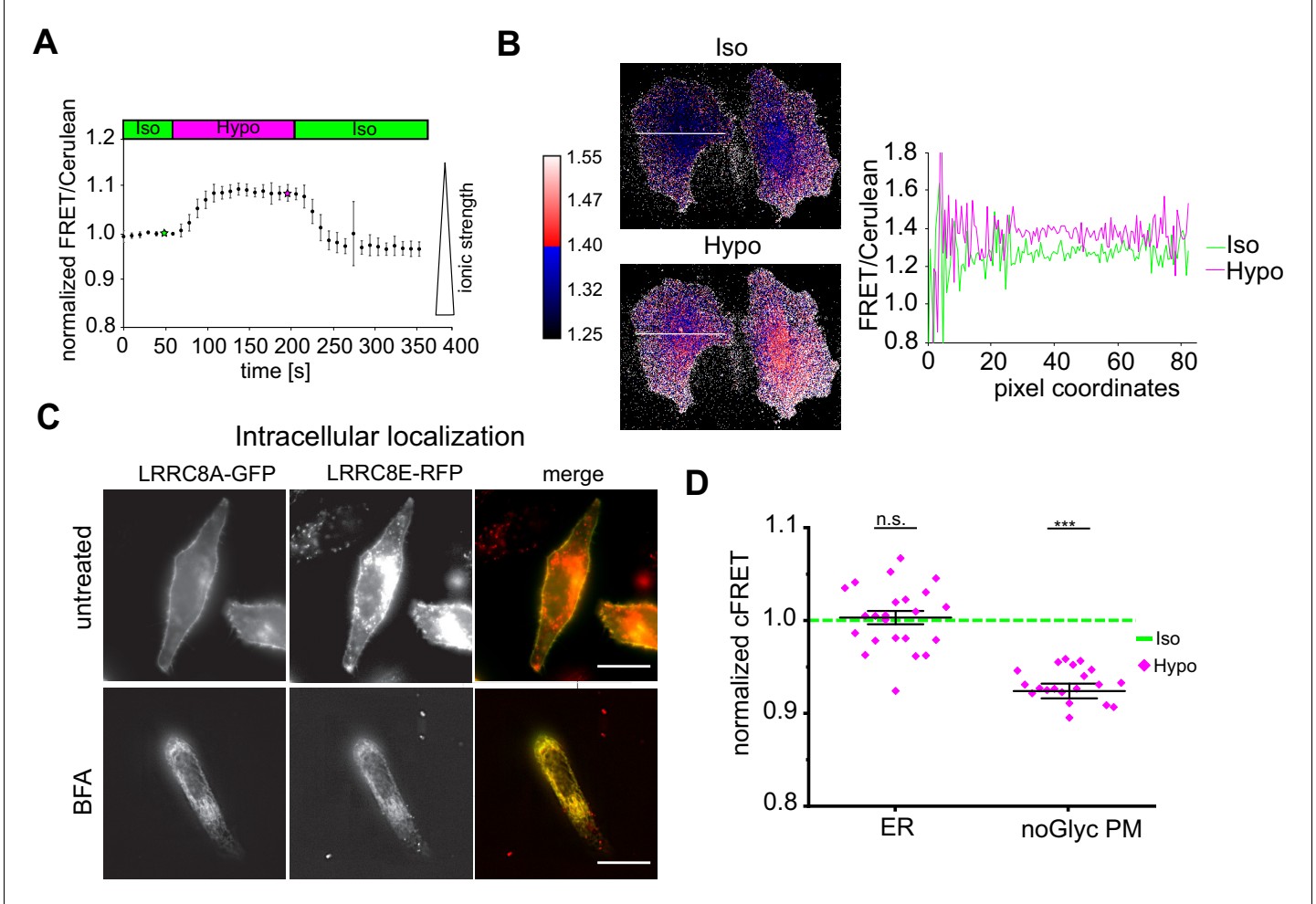

**Figure 2.** Hypotonicity reduces ionic strength over the whole cell, but does not activate intracellular VRACs. (**A**) Changes of ionic strength measured in HeLa cells expressing the ratiometric RD sensor during hypotonic treatment. FRET intensity was divided by Cerulean intensity and normalized to basal values, with increasing values corresponding to decreasing ionic strength. Data represent mean of 10 cells ± s.d. (**B**) Ratio maps of two representative cells in isotonic (green star in panel A) and hypotonic (magenta star in panel A) buffer. Right: intensity profile along the white line shown in the ratio maps. (**C**) Epifluorescence images of A-GFP and E-RFP in live cells without (top row) and with (bottom row) 5 µg/ml brefeldin-A (BFA) treatment. Images of BFA-treated cells were background-subtracted by rolling-ball algorithm (radius = 50 pixels) and contrast increased by an unsharp mask (sigma = 3, weight = 0.7). Scale bar, 20 µm. (**D**) Normalized cFRET of A-CFP/E-YFP trapped in the ER by presence of BFA during expression (ER; n = 16 dishes with 23 cells) and A$^{N66A,N83A}$-CFP/E-YFP without BFA (noGlyc PM; n = 7, 21 cells). Data represent individual cells (diamonds) and mean ± s.e.m. Statistics: ***p<0.0005; n.s., not significant, Student's *t*-test, comparing hypotonic with isotonic.

DOI: https://doi.org/10.7554/eLife.45421.006

The following video and source data are available for figure 2:

**Source data 1.** Statistics of hypotonicity-induced FRET changes.
DOI: https://doi.org/10.7554/eLife.45421.007

**Figure 2—video 1.** Reduced intracellular ionic strength upon hypo-osmotic treatment.
DOI: https://doi.org/10.7554/eLife.45421.008

potentiated VRAC activation, as judged by the reduction in cFRET (*Figure 3—figure supplement 2B,C*), as was previously shown for VRAC-mediated $I_{Cl,swell}$ in bovine aortic endothelial cells (*Levitan et al., 2000*). However, absence of cholesterol is unlikely to underlie plasma membrane localization as a prerequisite for VRAC activation, since the ER membrane usually has lower cholesterol concentrations than the plasma membrane.

Phospholipase C (PLC) activity was reported as a determinant of $I_{Cl,swell}$ activation, but contradictory results abound for the involvement of protein kinase C (PKC), activated by the PLC product

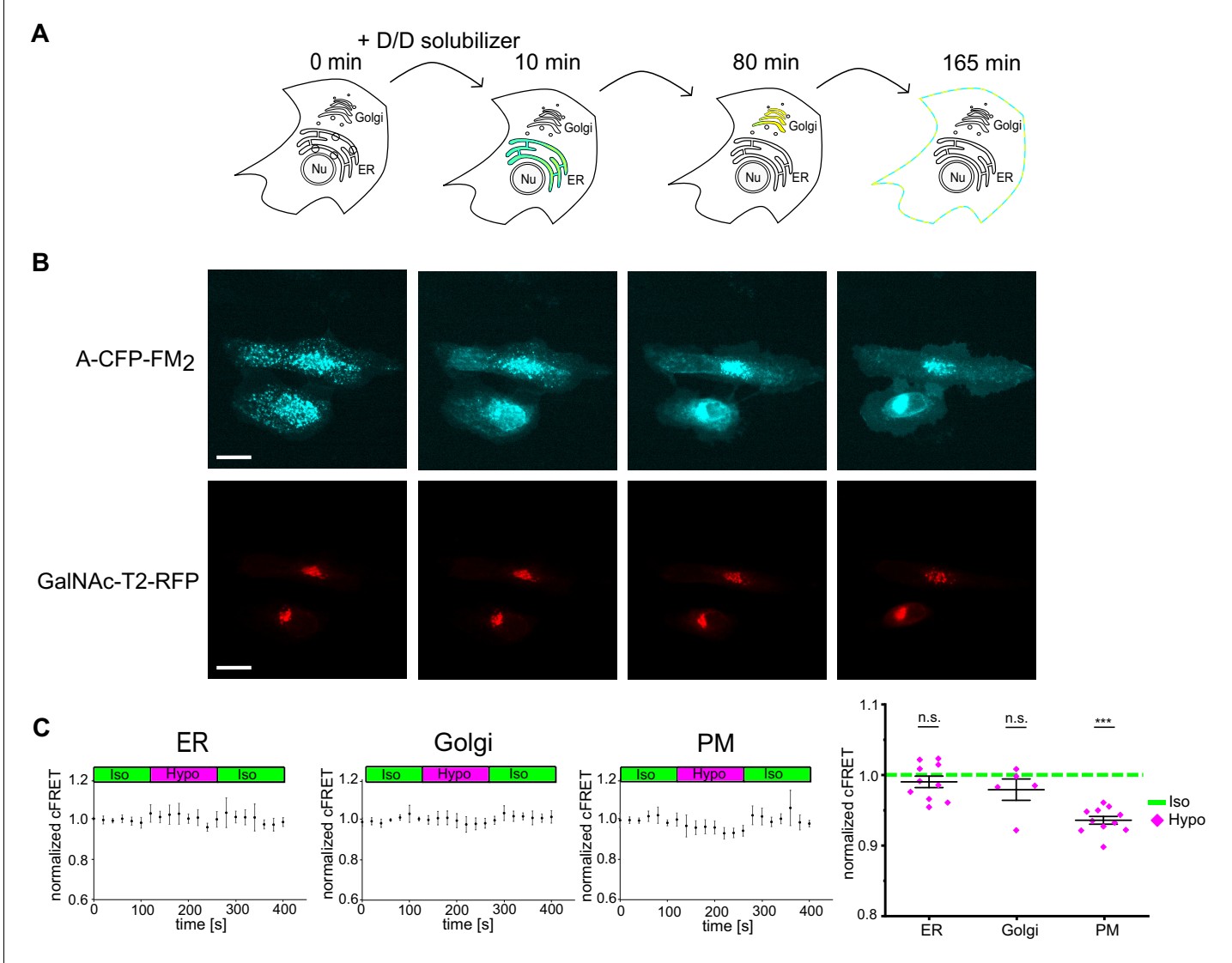

**Figure 3.** Plasma membrane localization is required for VRAC activation. (**A**) Scheme depicting the different subcellular localizations of VRACs using the reverse aggregation system. Upon addition of the D/D solubilizer, VRACs carrying FM dimerization domains disaggregate in the ER and travel through the Golgi complex to the plasma membrane. Indicated time points were deduced from experiments shown in (**B**) and (**Figure 3—video 1**). (**B**) Still images of HeLa cells expressing LRRC8A-CFP-FM$_2$ and GalNAcT2-RFP as Golgi marker from **Figure 3—video 1** at time points indicated in (**A**). Scale bar, 20 μm. (**C**) cFRET measured in HeLa cells expressing A-CFP-FM$_2$/E-YFP. VRACs localizing to the ER (n = 4, 10 cells), Golgi (n = 5, 26 cells) and plasma membrane (PM; n = 5, 11 cells) were measured at 10, 80 and 165 min after addition of D/D solubilizer. Data in time traces represent mean ± s.d. Right: average normalized cFRET in of the first seven time points in isotonic and last three time points in hypotonic buffer. Data represent individual cells (ER, PM) or FOVs (Golgi) and mean ± s.e.m. Statistics: ***p<0.0005; n.s., not significant, Student's *t*-test, comparing hypotonic with isotonic.

DOI: https://doi.org/10.7554/eLife.45421.009

The following video, source data, and figure supplements are available for figure 3:

**Source data 1.** Statistics of hypotonicity-induced FRET changes and cholesterol depletion.

DOI: https://doi.org/10.7554/eLife.45421.012

**Figure supplement 1.** Trafficking of LRRC8 complexes through the secretory pathway.

DOI: https://doi.org/10.7554/eLife.45421.010

**Figure supplement 2.** Modulation of VRAC activity by the actin cytoskeleton and cholesterol content.

DOI: https://doi.org/10.7554/eLife.45421.011

**Figure 3—video 1.** Trafficking of LRRC8 complexes through the secretory pathway.

DOI: https://doi.org/10.7554/eLife.45421.013

diacylglycol (DAG) is required (*Hermoso et al., 2004*; *Roman et al., 1998*; *Zholos et al., 2005*). PKC contains a C1 domain and can hence be activated by phorbol-12-myristate-13-acetate (PMA). In isotonic conditions, PMA evoked a robust decrease in cFRET in HeLa and HEK293 cells expressing A-CFP/A-YFP or A-CFP/E-YFP (*Figure 4A*), but not CFP-18aa-YFP (*Figure 4—figure supplement 1A*). This reflects gating of the LRRC8 complex, because subsequent inactivation of VRAC by treatment with hyper-osmotic solution restored basal FRET levels (*Figure 4—figure supplement 1B*). Like hypotonicity-driven VRAC activation, activation by PMA required cell surface localization, as VRACs released from aggregation were immune to PMA in the ER and Golgi, but they were activated at the plasma membrane (*Figure 4—figure supplement 1C*). Surprisingly, recording HEK293 KO cells expressing A-CFP/E-YFP in whole-cell voltage-clamp configuration, we could no longer

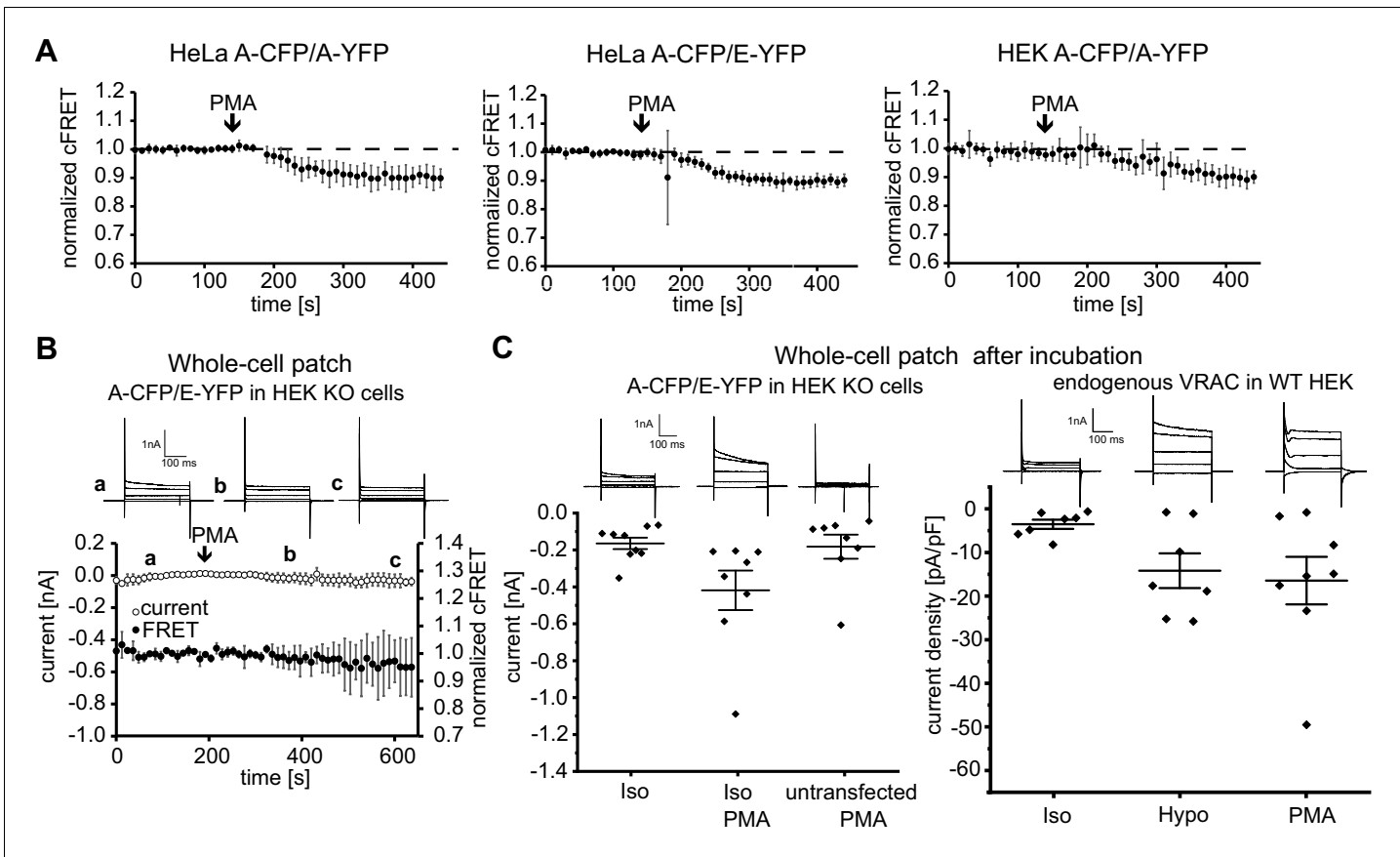

**Figure 4.** VRAC activation by PMA. (**A**) Addition PMA (1 µM, indicated by arrows) activated VRACs composed of A-CFP/A-YFP (left; 11 cells), A-CFP/E-YFP (middle; 12 cells) in HeLa cells or A-CFP/A-YFP in HEK293 cells (right; 12 cells). Data represent mean ± s.d. (**B**) Simultaneous measurements of whole-cell current at −80 mV (open circles) and normalized cFRET values (solid circles) in A-CFP/E-YFP-expressing HEK293 KO cells with addition of 1 µM PMA (indicated by arrow). Data represent mean of 3 cells ± s.d. Above, representative current traces from voltage-step protocols at the depicted time points (**a, b, c**). (**C**) Whole-cell currents at −80 mV of A-CFP/E-YFP-expressing (and untransfected as control, n = 8 cells) HEK293 KO cells (left) and current densities of wild-type HEK293 cells (right) with whole-cell configuration established after pre-incubation with indicated buffers: isotonic buffer without (n = 9) or with 1 µM PMA (n = 8) (left), isotonic (n = 7) and hypotonic (n = 7) buffers without and isotonic buffer with 1 µM PMA (n = 8) (right). Data represent individual cells (diamonds) and mean ± s.e.m. Above, representative current traces from voltage-step protocols for each condition.

DOI: https://doi.org/10.7554/eLife.45421.014

The following source data and figure supplements are available for figure 4:

**Source data 1.** Statistics of currents and FRET changes in presence of PMA or Gö6983.
DOI: https://doi.org/10.7554/eLife.45421.017
**Figure supplement 1.** PMA activates plasma membrane-localized VRAC.
DOI: https://doi.org/10.7554/eLife.45421.015
**Figure supplement 2.** Effects of Gö6983 on cFRET and VRAC currents.
DOI: https://doi.org/10.7554/eLife.45421.016

detect any FRET reduction, nor did PMA activate VRAC currents (*Figure 4B*). To test whether this discrepancy was due to the absence of endogenous LRRC8 proteins, or to the dialysis of the cell contents during whole-cell patch, we first incubated HEK293 KO cells expressing A-CFP/E-YFP (*Figure 4C*, left panel) and wild-type (WT) HEK293 (*Figure 4C*, right panel) with PMA for at least five minutes before membrane breakthrough. We then established the whole-cell configuration for immediate current measurements. Indeed, with this experimental design we observed a reliable increase in classical VRAC currents in HEK293 KO cells expressing A-CFP/E-YFP incubated with PMA. The increase was robust compared to cells in isotonic buffer alone, or untransfected HEK293 KO cells incubated with PMA (*Figure 4C*). This result demonstrates that the whole-cell voltage-clamp configuration affects signaling pathways involved in activation of VRAC in a time-dependent manner, an experimental caveat that can be overcome with our non-invasive FRET sensor.

## Protein kinase D activity, rather than ionic strength, determines VRAC activation

Using the PKC inhibitor Gö6983 not only prevented the cFRET reduction upon hypotonicity, it also gradually increased cFRET over the course of the experiment (*Figure 4—figure supplement 2A*). However, in the whole-cell patch-clamp configuration, the hypotonicity-induced current even per-sisted in isotonic buffer when supplemented with Gö6983 (*Figure 4—figure supplement 2B*, left panel). The cFRET of the same cells, on the other hand increased to high values (*Figure 4—figure supplement 2B*, middle panel). Again, we observed different results for the cell being recorded in whole-cell mode and neighboring non-dialyzed cells in the same field of view (FOV). For neighboring cells, cFRET changes matched those seen for HeLa cells expressing A-CFP/E-YFP (*Figure 4—figure supplement 2A,B*, right panel). Together, these data do not support a role for PKC in VRAC activation.

However, because DAG and PMA also recruit C1 domain-containing protein kinase D (PKD) to the plasma membrane (*Rozengurt, 2011*), we next assessed a possible involvement of PKD. The presence of the PKD inhibitor CRT 0066101 during hypotonic treatment abolished cFRET changes and strongly reduced VRAC currents in HEK293 KO cells expressing A-CFP/E-YFP (*Figure 5A*). To exclude the potential confounder of cell dialysis during voltage clamp, we also measured steady-state currents in cells that were incubated in hypotonic buffer before membrane breakthrough, with and without CRT 0066101 or Gö6983. Similar to incubation during the whole-cell patch-clamp experi-ment, the presence of CRT 0066101 readily diminished currents following breakthrough after pre-incubation in hypotonic buffer, whereas Gö6983 did not (*Figure 5B*). This suggests an essential role of DAG-activated PKD in hypotonicity-induced VRAC activation.

DAG is converted to phosphatidic acid by DAG kinases (DGKs), and so inhibition of DGKs by dio-ctanoylglycol (DOG) should lead to accumulation of DAG. According to our hypothesis concerning PKD signaling, following VRAC activation by hypotonicity, DOG should support persistent activity of VRAC even after returning to isotonic conditions. Indeed, in the presence of DOG, we observed a persistent VRAC current after switching from extracellular hypotonicity to normal, isotonic buffer for A-CFP/E-YFP expressed in HEK293 KO cells (*Figure 5C*, left panel, *Figure 5—figure supplement 1A*, left panel), as well as for endogenous VRAC in WT HEK293 cells (*Figure 5C*, right panel, *Figure 5—figure supplement 1A*, right panel). For the A-CFP/E-YFP VRAC expressed in either HEK293 KO cells in whole-cell patch clamp (*Figure 5—figure supplement 1A*, left panel) and non-dialyzed HeLa cells (*Figure 5D*, left panel, *Figure 5—figure supplement 1B*) the DGK inhibitor DOG impaired the recovery of cFRET to basal values when cells were returned to isotonicity after hypo-tonic treatment. While cFRET in HeLa cells without DOG recovered by >80% (recovery of normalized cFRET from 95.2 ± 2.5% in hypotonic buffer to 99.4 ± 4.0% in isotonic buffer for 47 cells), cFRET in DOG-treated HeLa cells recovered by less than 40% (from 89.6 ± 8.0% to 93.6 ± 5.9%, for 9 cells). In striking contrast, $\Gamma_i$ fully recovered to the normal levels irrespective of the presence of DOG (*Figure 2A*, *Figure 5D*, right panel). These results demonstrate that opening of VRACs is not directly coupled to $\Gamma_i$, but rather to the DAG pathway, likely through activation of PKD in HeLa and HEK293 cells.

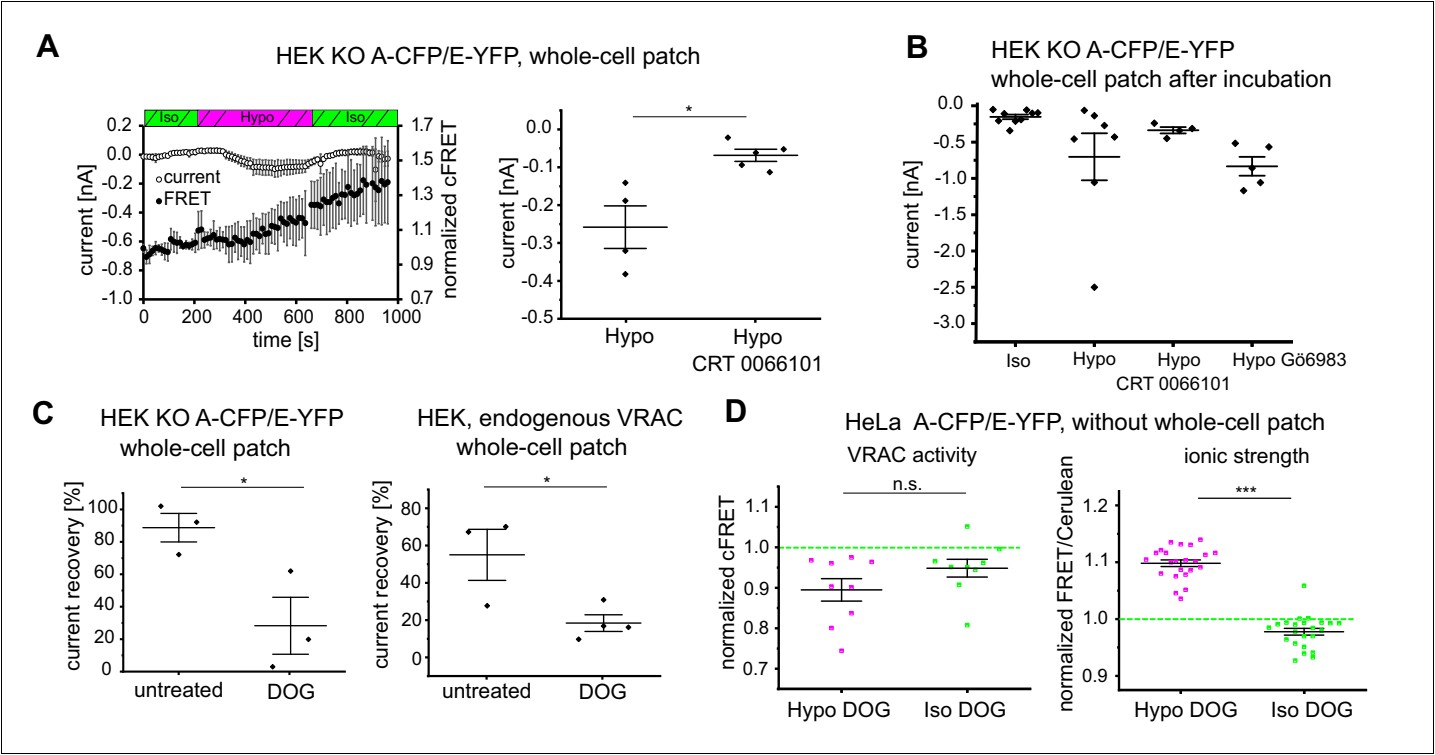

**Figure 5.** DAG and PKD as determinants of VRAC activity. (A) Left: simultaneous measurements of whole-cell current at −80 mV (open circles) and normalized cFRET values (solid circles) during buffer exchange experiment with A-CFP/E-YFP-expressing HEK293 KO cells, after 15 min pre-incubation with 5 µM CRT 0066101 in the presence of CRT 0066101. Data represent mean of 5 cells ± s.d. Right: average currents of the last five time points in hypotonicity with (data from left panel) and without (data from *Figure 1F*) CRT 0066101; data represent individual cells (diamonds) and mean ± s.e.m. (B) Whole-cell currents of HEK293 KO cells expressing A-CFP/E-YFP clamped at −80 mV after pre-incubation with indicated buffers: isotonic buffer (n = 9), hypotonic buffer (n = 7), hypotonic buffer with 5 µM CRT 0066101 (n = 4) or 1 µM Gö6983 (n = 5). Data represent individual cells (diamonds) and mean ± s.e.m. (C) Recovery of whole-cell currents in HEK293 cells expressing A-CFP/E-YFP (left) and wild-type HEK293 (right) at −80 mV to basal levels after switching from extracellular hypotonicity to isotonicity in the absence or presence of the DAG kinase inhibitor dioctanoylglycol (100 µM DOG). In HEK293 KO, n = 3 for untreated (from measurements in *Figure 1F*) and n = 3 for DOG-treated; in wild-type HEK293, n = 3 for untreated and n = 4 for DOG-treated. Data represent individual cells (diamonds) and mean ± s.e.m. (D) cFRET was measured for HeLa cells expressing either VRACs containing A-CFP/E-YFP (left, n = 7 dishes with 9 cells) or the RD sensor for ionic strength (right, n = 6, 23 cells) consecutively in isotonic buffer, hypotonic buffer supplemented with 100 µM DOG and isotonic buffer with DOG. Average cFRET of the last five time points per condition. Data represent individual cells and mean ± s.e.m. Statistics: *p<0.05; ***p<0.0005; n.s., not significant, Student's *t*-test.

DOI: https://doi.org/10.7554/eLife.45421.018

The following source data and figure supplement are available for figure 5:

**Source data 1.** Statistics of currents and FRET changes in presence of CRT 0066101 or DOG.

DOI: https://doi.org/10.7554/eLife.45421.020

**Figure supplement 1.** Dioctanoylglycol (DOG) reduces VRAC inactivation after induction by hypotonicity.

DOI: https://doi.org/10.7554/eLife.45421.019

## Discussion

Here we present an intra-complex FRET sensor to monitor VRAC activity in live cells with subcellular resolution. The relationship of the observed FRET changes to VRAC gating are clear from numerous common features (*Akita and Okada, 2014*; *König and Stauber, 2019*; *Pedersen et al., 2016*; *Strange et al., 2019*): (i) the reversible decrease in FRET matches reversible VRAC activation upon treatment with hypo-osmotic solutions, and with the same kinetics as simultaneously-measured whole-cell VRAC currents, (ii) the intensity of the FRET decrease correlates with the applied extracellular hypotonicity; (iii) the modulation of VRAC currents, for example by cholesterol depletion, is mirrored by the observed FRET alteration; (iv) after activation with PMA, FRET rises in hypertonicity as VRAC is inactivated by cell shrinkage. FRET sensors unrelated to VRAC are not affected by hypo-osmotic solutions, demonstrating that the observed changes with our FRET sensor are indeed

specific for VRAC activation. Moreover, our optical method to monitor VRAC activity has some clear advantages over electrophysiological approaches, including its application to study subcellular profiles of VRAC activity, and its non-invasive nature. The activation of low-conducting VRACs composed of only LRRC8A (*Deneka et al., 2018*; *Kefauver et al., 2018*; *Syeda et al., 2016*; *Yamada and Strange, 2018*) could also be detected by FRET. Importantly, the non-invasive, genetically-encoded FRET sensor enables the study of VRAC in connection with signaling pathways that are altered (presumably by dialysis of critical signaling molecules) by the whole-cell patch-clamp configuration. This way we could observe VRAC activation by PMA, which was otherwise blocked by whole-cell patch clamp of cells.

Cryo-EM structures of VRAC complexes suggest a certain flexibility of their C-terminal leucine-rich domains (LRRDs) on the intracellular side. Related conformational rearrangements might be related to channel gating (*Deneka et al., 2018*; *Kasuya et al., 2018*; *Kefauver et al., 2018*; *Kern et al., 2019*). In support of this idea, we observe a decrease in FRET between fluorophores at the C-termini of LRRC8 proteins upon VRAC activation, which may reflect a displacement of LRRDs or possibly a simple change in the average orientation of the fluorophores to each other. Thus, as has previously been shown for cyclic nucleotide-gated (CNG) channels (*Zheng and Zagotta, 2000*), slow gating of CLC chloride channels (*Bykova et al., 2006*), BK potassium channels (*Miranda et al., 2013*) and ionotropic glutamate receptors (*Zachariassen et al., 2016*), changes in FRET demonstrate movement of the cytosolic domains during VRAC activation.

VRACs were previously shown to be directly activated by low ionic strength in lipid bilayer reconstitution (*Syeda et al., 2016*). However, we find that reduced $\Gamma_i$ is neither sufficient to activate VRAC on intracellular compartments, nor is it indispensable to keep plasma membrane-localized VRACs active. When cells were treated with the DAG kinase inhibitor DOG, VRAC remained active in isotonic buffer after activation by hypotonicity, even though $\Gamma_i$ recovered to normal levels. The notion that ionic strength directly activates VRACs (*Sabirov et al., 2000*; *Voets et al., 1999*) has been controversial (*König and Stauber, 2019*; *Strange et al., 2019*). Several studies reported VRAC activation when isotonic fluid was injected into, or inactivation when fluid was withdrawn from the cells without altering the intracellular ionic strength (*Best and Brown, 2009*; *Cannon et al., 1998*; *Poletto Chaves and Varanda, 2008*; *Zhang and Lieberman, 1996*). Moreover, the extents of low ionic strength required for VRAC activation (*Cannon et al., 1998*; *Deneka et al., 2018*; *Nilius et al., 1998*; *Syeda et al., 2016*) are unlikely to occur in physiological conditions or during whole-cell current measurements routinely used to study hypotonicity-induced VRAC currents (*Strange et al., 2019*). Instead, a putative role for ionic strength in controlling the volume set point for VRAC activation (*Best and Brown, 2009*; *Cannon et al., 1998*; *Jackson et al., 1996*) would be consistent with our findings.

Rather than ionic strength, we observe a specific and potent action of the DAG-PKD pathway in VRAC activation. Application of PMA led to VRAC opening even under isotonic conditions and VRACs did not inactivate completely in isotonic buffer after activation by hypotonicity when the consumption of DAG by DAG kinase was inhibited. A role for DAG, which is generated by the activity of phospholipases C (PLCs), in the signaling for VRAC activation has been reported previously, for example (*Roman et al., 1998*; *Rudkouskaya et al., 2008*; *Zholos et al., 2005*). However, there are multiple contradictory reports on the involvement of DAG-activated PKCs. In some studies, PKCs were shown to be required for regulatory volume decrease (*Hermoso et al., 2004*), activation of swelling-induced VRAC currents (*Roman et al., 1998*) and ATP-induced VRAC-mediated glutamate release (*Hyzinski-García et al., 2014*; *Rudkouskaya et al., 2008*). These results are somewhat compromised by variations in the efficacy of pharmacological PKC inhibitors (*Haskew-Layton et al., 2005*; *Mongin and Kimelberg, 2005*). Other studies found no role for PKC despite a VRAC-activating effect of DAG (*Zholos et al., 2005*), or even reduction or prevention of hypotonicity-induced VRAC-mediated currents by pharmacological PKC inhibition (*Ben Soussia et al., 2012*; *Shlyonsky et al., 2011*; *von Weikersthal et al., 1999*). While our data speak against a core role for PKC in VRAC activation, they do not rule out a regulatory role for this kinase family. Instead, our experiments reveal that protein kinase D (PKD) activity is critically involved in hypotonicity-induced activation of VRAC. LRRC8 proteins possess several putative and some confirmed phosphorylation sites for various kinases in their intracellular loops and C-terminal domains (*Abascal and Zardoya, 2012*; *Mongin, 2016*). As opposed to direct phosphorylation of LRRC8 subunits, PKD may act on another, as yet unknown, third party in the process of VRAC activation.

The central role for DAG and PKD, however, does not explain the observed plasma membrane requirement for VRAC activation by hypotonicity as PKD is also recruited to endomembranes of the secretory pathway by DAG (*Baron and Malhotra, 2002*; *Fu and Rubin, 2011*). Plasma membrane-intrinsic properties such as membrane stiffness, the lipid composition, including the amount of cholesterol and different phosphatidylinositol phosphates (PIPs), or the transmembrane potential may play a role. Besides, further limiting factors are likely to be involved, potentially explaining why the overexpression of functional LRRC8 heteromers does not increase $I_{Cl,swell}$ above wild-type levels (*Voss et al., 2014*) and similar VRAC subunit expression levels displayed different activities in different cell types (*Okada et al., 2017*). Such limiting factors at the plasma membrane could include substrates of PKD. Although we also observed PMA-mediated activation of LRRC8A homomers, regulatory proteins could require particular VRAC subunit combinations to contribute to VRAC activation. Alternatively, cell type-specific factors may exist, explaining inconsistent findings on the signal transduction pathways impinging on VRAC activation. This could explain divergent results related to PKC in different cell lines (*Ben Soussia et al., 2012*; *Hermoso et al., 2004*; *Roman et al., 1998*; *von Weikersthal et al., 1999*; *Zholos et al., 2005*).

In summary, activation-dependent inter-subunit FRET in VRACs demonstrated movement of their C-termini and enabled subcellular monitoring of VRAC activity. Importantly, this noninvasive FRET sensor supersedes some aspects of electrophysiological measurements because it leaves intracellular signaling pathways untouched. Measurement of FRET can also be used for non-invasive structure-activity studies. The genetic encoding of this sensor also makes it highly amenable to studying VRAC activation during various cell physiological processes in situ.

# Materials and methods

**Key resources table**

| Reagent type (species) or resource | Designation | Source or reference | Identifiers | Additional information |
|---|---|---|---|---|
| Cell line (*Homo sapiens*) | HeLa | RRID: CVCL_0030 | ACC no. 57 | Obtained from Leibniz Forschungsinstitut DSMZ (Deutsche Sammlung von Mikroorganismen und Zellkulturen GmbH, Germany) |
| Cell line (*Homo sapiens*) | HEK293 | RRID: CVCL_0045 | ACC no. 305 | Obtained from Leibniz Forschungsinstitut DSMZ (Deutsche Sammlung von Mikroorganismen und Zellkulturen GmbH, Germany) |
| Cell line (*Homo sapiens*) | HEK293 KO (*LRRC8⁻/⁻*) | *Lutter et al., 2017* (PMID:28193731) | | Gift from T.J. Jentsch (MDC and FMP, Berlin, Germany) |
| Transfected construct | A-GFP (LRRC8A-GFP) | *Voss et al., 2014* (PMID: 24790029) | | Gift from T.J. Jentsch (MDC and FMP, Berlin, Germany) |
| Transfected construct | A-CFP (LRRC8A-CFP) | This paper | | LRRC8A-coding sequence subcloned from A-GFP (*Voss et al., 2014*) |
| Transfected construct | A-CFP-FM$_2$ (LRRC8A-CFP-FM$_2$) | This paper | | 2 FM domains (*Rollins et al., 2000*) cloned in LRRC8A-CFP with insertion sites generated by Q5-mutagenesis |

*Continued on next page*

*Continued*

| Reagent type (species) or resource | Designation | Source or reference | Identifiers | Additional information |
|---|---|---|---|---|
| Transfected construct | A$^{N66A,N83A}$-CFP (LRRC8A$^{N66A,N83A}$-CFP) | This paper | | LRRC8A-coding sequence subcloned from A$^{N66A,N83A}$-GFP (*Voss et al., 2014*; gift from T.J. Jentsch, MDC and FMP, Berlin, Germany) |
| Transfected construct | A-CFP (LRRC8A-Cerulean) | This paper | | LRRC8A-coding sequence subcloned from A-GFP (*Voss et al., 2014*) |
| Transfected construct | A-YFP (LRRC8A-YFP) | This paper | | LRRC8A-coding sequence subcloned from A-GFP (*Voss et al., 2014*) |
| Transfected construct | E-CFP (LRRC8E-CFP) | This paper | | LRRC8E-coding sequence subcloned from LRRC8E-GFP (*Voss et al., 2014*; gift from T.J. Jentsch, MDC and FMP, Berlin, Germany) |
| Transfected construct | E-YFP (LRRC8E-YFP) | This paper | | LRRC8E-coding sequence subcloned from LRRC8E-GFP (*Voss et al., 2014*) |
| Transfected construct | E-YFP (LRRC8E-Venus) | This paper | | LRRC8E-coding sequence subcloned from LRRC8E-GFP (*Voss et al., 2014*) |
| Transfected construct | E-RFP (LRRC8E-RFP) | This paper | | LRRC8E-coding sequence subcloned from LRRC8E-GFP (*Voss et al., 2014*) |
| Transfected construct | CFP-18AA-YFP | *Elder et al., 2009* (PMCID: PMC2706461) | | Gift from C.F. Kaminski (University of Cambridge, UK) |
| Transfected construct | GluA2-6Y-10C | *Zachariassen et al., 2016* (PMID:27313205) | | |
| Transfected construct | RD sensor (ionic strength sensor) | *Liu et al., 2017* (PMID: 28853549) | | Gift from B. Poolman and A.J. Boersma (University of Groningen, The Netherlands) |
| Transfected construct | ER-YFP (pEYFP-ER) | Clontech, TaKaRa | Catalog no. 6906–1 | |
| Transfected construct | GalNAc-T2-RFP | This paper | | Coding sequence of stalk region of GalNAc-T2 subcloned from GalNAc-T2-GFP (*Le Bot et al., 1998*; gift from I. Vernos, CRG, Barcelona, Spain) |
| Transfected construct | CD4-YFP | This paper | | hCD4-coding sequence subcloned from CD4-GFP (*Leisle et al., 2011*; PMID: 21527911) |

*Continued on next page*

*Continued*

| Reagent type (species) or resource | Designation | Source or reference | Identifiers | Additional information |
|---|---|---|---|---|
| Chemical compound, drug | Brefeldin-A; BFA | Sigma-Aldrich | Catalog no. B5936 | Ready-made solution of 10 mg/ml in DMSO-based solution, used at final concentration of 5 µg/ml |
| Chemical compound, drug | D/D solubilizer | Clontech, TaKaRa | Catalog no. 635054 | Obtained as solution, used at final concentration of 0.5 µM in growth medium |
| Chemical compound, drug | Latrunculin B; LatB | Sigma-Aldrich | Catalog no. 428020 | Stock prepared in DMSO, used at final concentration of 2 µM |
| Chemical compound, drug | Alexa Fluor 546-phalloidin | Thermo Fisher Scientific | Catalog no. A22283 | Diluted 1:1000 in PBS + 1% BSA |
| Chemical compound, drug | Methyl-β-cyclodextrin; MbCD | Sigma-Aldrich | Catalog no. C4555 | Dissolved in DMEM and sterile-filtered before use, used at final concentration of 5 mM |
| Chemical compound, drug | Filipin | Sigma-Aldrich | Catalog no. SAE0088 | Ready-made filipin complex solution of 5 mg/ml in DMSO-based solution, used at final concentration of 125 µg/ml in PBS |
| Chemical compound, drug | Phorbol-12-myristat-13-acetat; PMA | Tocris, Bio-Techne | Catalog no. 1201/1 | Stock prepared in DMSO, used at final concentration of 1 µM |
| Chemical compound, drug | Gö6983 | Abcam | Catalog no. ab144414 | Stock prepared in DMSO, used at final concentration of 1 µM |
| Chemical compound, drug | CRT 0066101 | Tocris, Bio-Techne | Catalog no. 4975/10 | Stock prepared in $H_2O$, used at final concentration of 5 µM |
| Chemical compound, drug | Dioctanoylglycol; DOG | Tocris, Bio-Techne | Catalog no. 0484 | Stock prepared in DMSO, used at final concentration of 100 µM |
| Software, algorithm | Fiji | *Schindelin et al., 2012* (PMID: 22743772) | | |
| Software, algorithm | PixFRET plugin | *Feige et al., 2005* (PMID: 16208719) | | |
| Software, algorithm | µManager | *Edelstein et al., 2014* (PMID: 25606571) | | |
| Software, algorithm | PySerial library | https://pythonhosted.org/pyserial/index.html | | |

## Expression constructs

Expression plasmids for human LRRC8A-GFP, LRRC8A[N66A/N83A]-GFP and LRRC8E-RFP (*Voss et al., 2014*) were kindly provided by T.J. Jentsch. To generate FRET constructs, the coding sequences were subcloned with *EcoRI* and *KpnI* into pECFP-N1 and pEYFP-N1, resulting in 12-amino acid linkers LVPRARDPPVAT for LRRC8A or WVPRARDPPVAT for LRRC8E. For electrophysiological experiments, CFP and YFP were replaced with Cerulean and Venus by adding *AgeI* and *NotI* sites and insertion into the respective CFP- or YFP-tagged versions without altering the linker region. Cerulean and Venus are also referred to as CFP and YFP throughout. For expression of CD4-YFP, human CD4 was subcloned from CD4-GFP (*Leisle et al., 2011*) into pEYFP-N3. For the generation of A-CFP-FM₂, two FM domains (*Rollins et al., 2000*) were inserted into *XbaI-SpeI* restriction sites 3' of A-CFP

that were generated using the Q5 sited directed mutagenesis kit (New England Biolabs) with forward primer 5´ATCACTAGTAGCGGCCGCGACTCTAGA and reverse primer 5´ATCTCTAGACTTG TACAGCTCGTCCATGCC. The FRET-based RD sensor for ionic strength (*Liu et al., 2017*) was kindly provided by B. Poolman and A.J. Boersma, CFP-18AA-YFP (*Elder et al., 2009*) by C.F. Kaminski. The glutamate receptor construct GluA2-6Y-10C has been described previously (*Zachariassen et al., 2016*). For expression of GalNAcT2-RFP, the stalk region of N-acetylgalactosaminyltransferase 2 (GalNAcT2) was subcloned from pGalNAcT2-GFP (*Le Bot et al., 1998*) into pmRFP-N1. For expression of ER-localized YFP (ER-YFP), we used the plasmid pEYFP-ER (Clontech).

## Cell lines

HeLa (RRID: CVCL_0030) and HEK293 (RRID: CVCL_0045) cells were obtained from Leibniz Forschungsinstitut DSMZ and regularly tested for mycoplasma contamination. *LRRC8*[-/-] HEK293 (HEK293 KO) cells deficient in all five LRRC8 subunits (*Lutter et al., 2017*) were kindly provided by T.J. Jentsch. Cells were grown in DMEM (Pan-Biotech) supplemented with 10% fetal calf serum at 37˚C in 5% $CO_2$. For imaging experiments without simultaneous electrophysiology, cells were plated in 35 mm glass bottom dishes (MatTek), coated with poly-L-lysine 0.01% solution (Sigma-Aldrich) for HEK293 cells. For electrophysiology, cells were plated on poly-L-lysine-coated 25 mm coverslips. Cells were transfected with FuGENE 6 (Promega) according to the supplier's manual. For co-expression, constructs were co-transfected at equimolar ratios.

## Drug treatment

Brefeldin-A (BFA, Sigma-Aldrich, 10 mg/ml in DMSO) was added at 5 µg/ml to culture medium during transfection. To depolymerize the actin cytoskeleton, 2 µM latrunculin B (Sigma-Aldrich, dissolved in DMSO) was added to the growth medium for 1 hr in normal growth conditions. For cholesterol depletion, 5 mM methyl-β-cyclodextrin (MbCD, Sigma-Aldrich), dissolved in DMEM and stirred for 30 min at RT before sterile filtration, was applied for 1 hr in normal growth conditions. 1 µM phorbol-12-myristat-13-acetat (PMA, Bio-Techne, dissolved in DMSO), 1 µM Gö6983 (Abcam, DMSO), 5 µM CRT 0066101 (Bio-Techne, $H_2O$) and 100 µM dioctanoylglycol (DOG, Bio-Techne, DMSO) were added during or before measurements as indicated.

## Imaging of intracellular localization, actin cytoskeleton and cholesterol staining

Images of the intracellular localization of LRRC8A-GFP/LRRC8E-RFP and of LRRC8A-CFP-$FM_2$ with the organelle markers ER-YFP, GalNAc-T2-RFP and CD4-YFP were acquired with a 64x/1.4 oil objective at a Leica Dmi8 microscope equipped with a Hamamatsu OcraFlash4.0 controlled by LAS X. Images of the actin cytoskeleton and cholesterol content were acquired with a 20x/0.7 dry objective at the same microscope. For cholesterol staining, cells were washed with PBS, fixed with 4% paraformaldehyde (PFA) for 15 min at room temperature, washed with PBS and incubated with DMEM +10% FCS for 10 min at 37˚C, and, after washing with PBS, stained with 125 µg/ml filipin (Sigma-Aldrich, dissolved in DMSO and diluted in PBS) for 45 min at room temperature. After washing with PBS, images were acquired with a DAPI filter set (EX: 360/40, DC: 400, EM: 425 lp). To stain the actin cytoskeleton, PFA-fixed cells (as above) were permeabilized with 0.1% Triton-X100 in PBS + 1% BSA for 5 min at room temperature and incubated with Alexa Fluor 546-phalloidin (Thermo Fisher Scientific) diluted 1:1000 in PBS + 1% BSA at room temperature for 1 hr, washed with PBS, air dried and mounted on glass cover slides for imaging with a rhodamine filter set (EX:540/23, DC:580,EM:590 lp). Cells co-expressing LRRC8A-CFP-$FM_2$ with organelle markers were fixed and mounted as described above 24 hr after transfection and incubation with D/D solubilizer for the indicated times. Images where acquired with filter sets for CFP (EX:436/20, DC:455 EM:480/40), YFP (EX:500/20, DC:515, EM:535/30) and rhodamine (as above).

## Acceptor photo-bleaching, sensitized-emission and ratiometric FRET assays

FRET experiments were performed on a high-speed setup of Leica Microsystems (Dmi6000B stage, 63x/1.4 objective, high speed external Leica filter wheels with Leica FRET set filters (11522073), EL6000 light source, DFC360 FX camera, controlled by Las AF). All experiments were executed at

room temperature. Isotonic imaging buffer (340 mOsm) contained (in mM): 150 NaCl, 6 KCl, 1 MgCl$_2$, 1.5 CaCl$_2$, 10 glucose,10 HEPES, pH 7.4. NaCl was adjusted for osmolarity titration in hypotonicity; standard hypotonic buffer (250 mOsm) contained 105 mM NaCl. Hypertonic buffers were as isotonic buffer supplemented with 60 mM (for 400 mOsm) or 160 mM (for 500 mOsm) mannitol.

Acceptor bleaching experiments were conducted on field-of views (FOVs) with at least two cells expressing both fluorescent proteins. An image in both CFP and YFP channels was taken per time point. To bleach only part of the FOV, the field diaphragm was closed until control cells were no longer illuminated. Cells were bleached with 515–560 nm light at full intensity (Leica filter cube N2.1) for 2.5–5 min per time point until YFP signals reached below 10% of initial intensity. Intensities of CFP and YFP for respective time points were measured in Fiji in background-subtracted images and the percentage change of CFP intensity calculated for bleached and control cells.

Sensitized-emission FRET (seFRET) images were recorded with the same settings for donor, acceptor and FRET channels (8 × 8 binning, 100 ms exposure, gain 1) every 10–20 s. Images were processed with Fiji (*Schindelin et al., 2012*). cFRET maps were calculated with PixFRET plugin (*Feige et al., 2005*) (threshold set to 1, Gaussian blur to 2) with a self-written macro to process movies according to following equation (*Jiang and Sorkin, 2002*)

$$cFRET = \frac{IDA - IDD * \beta - IAA * \gamma}{IAA}$$

with emission intensities of I$^{DA}$ (FRET channel), I$^{DD}$ (donor channel), I$^{AA}$ (acceptor channel). Correction factors ($\beta$ = bleed through of donor emission; $\gamma$ = cross excitation of acceptor by donor excitation) were calculated from acceptor- and donor-only samples that were measured on three different days with at least three FOVs per day. cFRET maps were measured by manually-drawn regions of interest (ROIs) over whole cells, excluding intracellular aggregates that were oversaturated in any of the three channels. Because of the variability in absolute FRET values between individual cells, cFRET values of individual cells were normalized to their mean cFRET in isotonic buffer.

For monitoring ionic strength in living cells, we used the ratiometric FRET-based RD sensor (*Liu et al., 2017*). Sequential images of the CFP and FRET channels were acquired using the same filters as for seFRET experiments every 10 s. Images were acquired with 2 × 2 binning, 500 ms exposure, gain 3.0 and intensity four for hypotonicity experiments and with 8 × 8 binning, 127 ms exposure, gain 3.0 and intensity two for DOG experiments. Ratio maps of I_FRET/I_Cerulean were created from background-subtracted images in Fiji and mean ratio values were taken from hand-drawn ROIs of individual cells. To account for ionic strength differences between individual cells, ratios were normalized to baseline values in isotonic buffer.

## Electrophysiology and simultaneous seFRET measurements

For whole-cell patch-clamp recordings, the isotonic bath solution was composed as follows (in mM): 150 NaCl, 6 KCl, 1 MgCl$_2$, 1.5 CaCl$_2$, 10 glucose, and 10 HEPES, pH 7.4 with NaOH (320 mOsm). We used an intracellular (pipette) solution with the following composition (in mM): 40 CsCl, 100 Cs-methanesulfonate, 1 MgCl$_2$, 1.9 CaCl$_2$, 5 EGTA, 4 Na$_2$ATP, and 10 HEPES, with pH adjusted to 7.2 with CsOH (290 mOsm). For VRAC activation, cells were perfused with drugs and/or a hypotonic buffer containing (in mM) 105 NaCl, 6 CsCl, 1 MgCl$_2$, 1.5 CaCl$_2$, 10 glucose, 10 HEPES, pH 7.4 with NaOH (240 mOsm). The osmolarities of all solutions were measured with a vapor pressure osmometer (VAPRO 5600, Elitech). Pipettes had a resistance of 3–5 MΩ when filled with intracellular solution, and we used ISO-type pipette holders (G23 Instruments) to minimize pipette drift. For cells overexpressing exogenous, fluorescently-labelled VRAC constructs, the success rate for obtaining a gigaseal and whole-cell configuration were <20%, compared to >50% for wild-type untransfected HEK293 cells. Whole-cell recordings from cells overexpressing VRAC were also less stable. For normalizing current to cell capacitance, we used the capacitance from the manual compensation of transients from the Axopatch 200B amplifier (Molecular Devices). After whole-cell configuration was obtained, cells were held at –30 mV and the currents were recorded using Axograph X (Axograph Scientific) via an Instrutech ITC-18 D-A interface (HEKA Elektronik). The standard protocol for VRAC current recordings consisted of 9 episodes of a 0.5 s step to –80 mV every 12 s, followed by a 3 s voltage step protocol (300 ms steps to –40, 0, 40, 80, 100 mV in turn). This protocol was repeated throughout the recording. The voltage-step protocol confirmed VRAC-typical properties of outward rectification and depolarization-dependent inactivation for LRRC8A/E-containing VRAC (*König and*

*Stauber, 2019*; *Ullrich et al., 2016*; *Voss et al., 2014*). Excitation by 445 and 514 nm diode lasers (iChrome MLE, Toptica Photonics) for CFP and YFP was directed through a manual total internal reflection fluorescence (TIRF) input to an Olympus IX81 microscope. We used a 40 × Olympus objective (NA 0.6) for all recordings. For simultaneous ratiometric FRET recordings, filters specific to donor and acceptor excitation and emission wavelengths were alternated using the fluorescence turret. We controlled the microscope turret control with an in-house script written in PYTHON. Briefly, the turret was synchronized to the electrophysiology acquisition computer using voltage output from the digitizer via an Arduino microcontroller to a reverse-engineered serial connection (using the PySerial library; https://pythonhosted.org/pyserial/index.html) from a second computer to the IX81 microscope. Fluorescence intensities in response to 445 and 514 nm excitation were recorded sequentially with 100 ms exposure time and 2 × 2 binning on a Prime 95B CMOS camera (Photometrics) fitted to Cairn Optosplit-II. CFP and YFP emission were split with a T495lpxr dichroic mirror, and recorded on the same frame after passing through ET520/40 m (YFP) and ET470/24 m (CFP) filters (all mounted within the Optosplit, all Chroma). Exposures were timed to precede the –80 mV voltage steps. Laser emission and camera exposure were triggered in hardware directly from the digitizer. Images were recorded with Micromanager (*Edelstein et al., 2014*) and analyzed with FIJI as described above.

## Reverse aggregation system

A-CFP-FM$_2$- or A-CFP-FM$_2$/E-YFP-expressing cells were measured 24 hr post-transfection. Aggregates were released by adding 0.5 μM D/D solubilizer (TaKaRa) to the growth medium and incubation for 10, 80 or 165 min at 37°C in 5% $CO_2$ before seFRET measurements. All buffers used during the measurement contained 0.5 μM D/D solubilizer. Mean cFRET was acquired from manually-drawn ROIs over the whole cell for ER- and plasma membrane-localized VRAC. For plasma membrane, only cells with clear plasma membrane localization were selected by visual inspection. To determine the cFRET of Golgi-localized VRAC, masks were created on CFP images by applying a threshold over the high-intensity, juxta-nuclear regions and applying these masks on the respective cFRET maps.

## Statistical analysis

Data are represented as mean of individual cells ± standard deviation (s.d.) or mean of n (number of independent experiments) ± standard error of the mean (s.e.m.) as indicated. P values were determined by two-tailed Student's *t*-test and are indicated in all figures according to convention: n. s. = not significant, $*p \leq 0.05$, $**p \leq 0.005$ and $***p \leq 0.0005$.

# Acknowledgements

We thank Jonas Engelke for pilot experiments and help with the programming, Helge Ewers for providing experimental equipment, Iva Lučić for discussions and Arnold Boersma, Francesca Bottanelli, Thomas Jentsch, Clemens Kaminski, Bert Poolman and Isabelle Vernos for plasmids and cell lines.

# Additional information

## Funding

| Funder | Grant reference number | Author |
| --- | --- | --- |
| Bundesministerium für Bildung und Forschung | 031A314 | Tobias Stauber |

The funders had no role in study design, data collection and interpretation, or the decision to submit the work for publication.

## Author contributions

Benjamin König, Conceptualization, Data curation, Software, Formal analysis, Investigation, Visualization, Methodology, Writing—original draft, Writing—review and editing; Yuchen Hao, Investigation, Visualization, Methodology; Sophia Schwartz, Investigation, Visualization; Andrew JR Plested, Conceptualization, Resources, Software, Supervision, Visualization, Methodology, Writing—

review and editing; Tobias Stauber, Conceptualization, Resources, Supervision, Funding acquisition, Visualization, Methodology, Writing—original draft, Project administration, Writing—review and editing

**Author ORCIDs**
Andrew JR Plested  https://orcid.org/0000-0001-6062-0832
Tobias Stauber  https://orcid.org/0000-0003-0727-6109

**Decision letter and Author response**
Decision letter https://doi.org/10.7554/eLife.45421.023
Author response https://doi.org/10.7554/eLife.45421.024

## Additional files

**Supplementary files**
• Transparent reporting form
DOI: https://doi.org/10.7554/eLife.45421.021

**Data availability**
All data generated or analysed during this study are included in the manuscript and supporting files.

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
