## [Decision Letter]

[Editors’ note: the authors were asked to provide a plan for revisions before the editors issued a final decision. What follows is the editors’ letter requesting such plan.]

Thank you for sending your article entitled "FRET changes by C-terminal movement reveal VRAC activation by plasma membrane PKC signaling rather than ionic strength" for peer review at *eLife*. Your article is being evaluated by three peer reviewers, one of whom is a member of our Board of Reviewing Editors, and the evaluation is being overseen by a Reviewing Editor and Richard Aldrich as the Senior Editor.

Given the list of essential revisions, including new experiments, the editors and reviewers invite you to respond within the next two weeks with an action plan and timetable for the completion of the additional work. We plan to share your responses with the reviewers and then issue a binding recommendation.

The three reviewers agree that the main concern in this paper is that there isn't a validation by electrophysiology of the FRET data. It is essential that you perform simultaneous experiments measuring FRET and current in response to the activating stimuli. If this is not within your technical possibilities, at least the same kind of experiments to measure FRET should be repeated by electrophysiology. It is also essential that the FRET and imaging data is presented in a clearer manner and that the PKC FRET experiments be also validate by electrophysiology. Please see the reviewers report below for a more detailed list of comments.

Reviewer #1:

This paper by the Stauber group presents results from experiments in which they have measured a FRET signal between an attached fluorescent protein pair in the LRRC8A or E proteins which form the volume-regulated VRAC channels. The measurements show that the FRET signal is modulated by application of hypotonic solutions. The authors assert that the signal is related to activation of the VRAC channel and present further experimental evidence that activation is also mediated by activation of PKC phosphorylation. The experimental methods are adequate and in general the conclusions are supported by the evidence.

There are a few points that need to be presented in a clearer manner. In particular, while there is evidence that the FRET signal is specific of LRRC8, the authors cite a couple of reviews as evidence that the FRET signal correlates with the time course of activation of VRAC by the same stimulus. However, the evidence in those references shows activation of VRAC by voltage clamp stimuli occurs in an ms time scale and no data is shown on the time course of responses to a hypertonic stimulus. The authors need to show a parallel experiment showing the magnitude of the VRAC current produced by LRRC8A/A and LRRC8A/E in their particular expression system.

Voltage-clamp experiments show that VRAC currents inactivate in response to sustained applied stimulus. It is important to discuss the sustained nature of the FRET signal if it is related to channel activation. Is it thought that there are distinct gates for each modality of channel activation?

Reviewer #2:

For editors

The authors developed a highly interesting assay to monitor activity of LRRC8 mediated VRAC channels using a purely optical FRET based assay. Such an assay could be useful for structure-function studies, as well as for studies of VRAC function in physiological settings. However, a major criticism is that the assay has not been validated by parallel electrophysiological measurements. In case the authors can provide such validation I would be in favor of publication.

For authors

The manuscript by König et al., reports a method to monitor VRAC/LRRC8 volume sensitive channel activity in a time- and space-resolved manner by measuring intra-complex FRET between C-terminally fused fluorescent proteins. Two major aims are pursued: first, to demonstrate the flexibility of the leucine-rich-repeat domains (LRRDs) and their involvement in the gating of the channel; second, to provide evidence that low intracellular ionic strength is not the physiologically relevant parameter for the opening of VRACs. Rather, the authors propose that activation under hypotonic stimulation is coupled to PKC activity.

In addition, the authors use the assay to show that LRRC8 mediated VRAC is not activatable in endomembranes, but only in the plasma membrane.

The crucial questions driving the work are laudable and potentially very important for the field. In particular, the method of introducing an intramolecular FRET sensor of subunit distance is highly innovative and potentially very useful. The possibility to monitor LRRC8 activity with a purely optical assay is very intriguing.

However, several technical controls are necessary to fully validate the assay. Most importantly, it is mandatory that the assay itself, and other results, are corroborated by electrophysiological data.

All the experiments should be complemented with electrophysiological recordings. In principle, it would be most convincing if the FRET assay could be performed on the same cell that is patch-clamped to verify that FRET changes reflect current activation. Since this maybe technically too demanding, it is at least necessary to validate the constructs used by electrophysiological recordings, under the same conditions as the FRET assay. Attaching fluorescent proteins to the C-terminus might alter the "volume-sensitivity" by itself. In fact, the addition of fluorescent proteins to the C-terminus resulted in constitutive anion channel activity in *Xenopus* oocytes (Gaitán-Peñas et al., 2016).

The authors propose a central role for PKC in the gating of the channel and indicate that its pharmacological inhibition prevented hypotonicity-driven VRAC opening. Again, this result should be verified by electrophysiological recordings employing the same constructs, and constructs without tag – a quantitative comparison of the current density with and without application of PMA / Gö6983 / staurosporine is necessary.

*Reviewer #3:*

Volume-regulated anion channels (VRACs) play a central role in the modulation of cell volume in animals, by enabling the flow of chloride ions and larger osmolytes out of the cell upon activation by hypotonic conditions, and therefore counteracting cell-swelling. These channels are heterohexamers composed of the obligatory subunit LRRC8A and additional LRRC8-B-E subunits. The mechanism of activation of VRAC in not understood, but it has been proposed to be related to conformational changes at its C-terminus, and has been shown to be triggered by reduced ionic strength in a reconstituted system. In the present manuscript, Benjamin Koenig and collaborators construct a FRET-based VRAC-activity sensor by fusing the C-termini of different subunits with either of two fluorescence proteins capable of FRET. The authors show changes in FRET in response to hypotonic media that are reversible upon return to isotonic conditions, with kinetics and pharmacology that is consistent with the signal originating from VRAC activation. They also show that hypotonicity-driven FRET changes occur only for channels in the plasma membrane, whereas those located at intracellular compartments (ER and golgi) don't exhibit changes in FRET. They demonstrate that the requirement for activity of localization in the plasma membrane does not involve differences in cholesterol concentration or the integrity of the actin cytoskeleton. In contrast, the authors find that stimulation of PKC activity leads to prominent FRET changes similar to those produced by hypertonic media, whereas inhibition of PKC prevents FRET changes in response to hypotonic media. The authors have developed a set of clever tools to follow VRAC activity without requiring electrophysiological measurements, and provide relevant information on the cell-physiological conditions that enable VRAC activity and on the role of the C-terminus in channel activation. However, I think the manuscript would be much stronger if the authors provided more direct evidence that the FRET changes are indeed connected with channel activation. In addition, they should address some apparent inconsistencies within their observations and also relative to published data, and provide additional information on some experiments.

I have the following specific concerns:

1) All conclusions in the manuscript depend on the assumption that the observed FRET changes are reflecting channel activation. However, the evidence supporting this is not direct, but rather depends on correlative observations of the effects of non-specific treatments that could also affect other processes in the cell. It is possible that the FRET changes reflect a conformational change that is independent or at least partially independent of activation. This issue is particularly important given that the role of the C-terminus for activation has not been well established. I think the manuscript would be significantly stronger if the authors could provide more direct evidence for this, by performing electrophysiological measurements in combination with FRET measurements, or by using additional specific VRAC inhibitors or performing experiments with loss-of-function VRAC subunits.

2) One of the main observations of the paper is that localization in the plasma membrane is required for the observed FRET changes and channel activation. However, there is a significant amount of intracellularly located LRRC8 subunits even without treatment with BFA (see LRRC8E-RFP in Figure 2C, for example), and the hypotonicity-dependent FRET signals shown in Figure 1 appear to originate from the entire cell (see Figure 1D). The authors should address this apparent contradiction, and provide detailed information on the way they performed their analysis to measure signals mostly originating from the plasma membrane. The observation that LRRC8E appears to be more heavily retained in the ER that LRRC8A (see Figure 2C) seems to counter the observation that the 8A/8E ratio does not correlate with the observed changes in FRET, except if the majority of the measured FRET signal originates from the plasma membrane. The authors should discuss these issues.

3) The authors should discuss the apparent contradiction of their data with the observations in a reconstituted system where decreased ionic strength is sufficient to activate VRAC channels.

4) From the images in Figure 3B, it is impossible for me to determine which channels are localized at the ER vs golgi, even though the authors used a marker for golgi expression. The authors should use ER, golgi and plasma membrane markers, and provide a semi-quantitative analysis to demonstrate the location of the different channel populations, and provide a more detailed description of how they selected their ROIs for performing group analysis of channels in each of the cellular compartments.

5) In Figure 1—figure supplement 1, the authors should show the net FRET rather than the normalized values, to allow readers to see how much variability there was in FRET between the controls and the VRACs.

6) The authors should show whether the treatments with LatB or MbCD affected net FRET levels, rather than just the normalized values.

7) The Abstract, Title and the overall text seem to imply that PKC activation is connected with the physiological activation of VRACs upon exposure to hypotonic solutions. However, the presented data does not really establish a link between VRAC stimulation by PKC and changes in extracellular osmolarity. It might very well be that PKC-stimulated VRACs are indeed activated by ionic strength, and that this sensitivity is obscured by PKC inhibition. The authors should provide a more lengthy discussion on how PKC might be affecting channel activity, even if based only on speculation.

---

## [Author Response]

[Editors' note: the authors’ plan for revisions was approved and the authors made a formal revised submission.]

Given the list of essential revisions, including new experiments, the editors and reviewers invite you to respond within the next two weeks with an action plan and timetable for the completion of the additional work. We plan to share your responses with the reviewers and then issue a binding recommendation.The three reviewers agree that the main concern in this paper is that there isn't a validation by electrophysiology of the FRET data. It is essential that you perform simultaneous experiments measuring FRET and current in response to the activating stimuli. If this is not within your technical possibilities, at least the same kind of experiments to measure FRET should be repeated by electrophysiology. It is also essential that the FRET and imaging data is presented in a clearer manner and that the PKC FRET experiments be also validate by electrophysiology. Please see the reviewers report below for a more detailed list of comments.

As we describe in the point-by-point response below, we have aimed to address all these points. We have conducted patch-clamp fluorometry to measure FRET and currents simultaneously, demonstrating the relation of the change in FRET signal with VRAC activation. We improved the presentation of our data and experimental methods. Furthermore, we have performed electrophysiological experiments and simultaneous FRET/current measurements to investigate the activation mechanism of VRAC. By doing so, we have confirmed the role of diacylglycerol. Instead of PKC, we were able to identify a role for PKD in VRAC activation by using the combined power of electrophysiology and our FRET sensor. In addition, we have surprisingly found whole-cell configuration-related artifacts that have would have remained uncovered without our optical tool.

Reviewer #1:This paper by the Stauber group presents results from experiments in which they have measured a FRET signal between an attached fluorescent protein pair in the LRRC8A or E proteins which form the volume-regulated VRAC channels. The measurements show that the FRET signal is modulated by application of hypotonic solutions. The authors assert that the signal is related to activation of the VRAC channel and present further experimental evidence that activation is also mediated by activation of PKC phosphorylation. The experimental methods are adequate and in general the conclusions are supported by the evidence.

We thank the reviewer for these positive comments.

There are a few points that need to be presented in a clearer manner. In particular, while there is evidence that the FRET signal is specific of LRRC8, the authors cite a couple of reviews as evidence that the FRET signal correlates with the time course of activation of VRAC by the same stimulus. However, the evidence in those references shows activation of VRAC by voltage clamp stimuli occurs in an ms time scale and no data is shown on the time course of responses to a hypertonic stimulus. The authors need to show a parallel experiment showing the magnitude of the VRAC current produced by LRRC8A/A and LRRC8A/E in their particular expression system.

Following the reviewers’ suggestions, we have now performed patch-clamp fluorometry to simultaneously measure FRET and currents of LRRC8A-CFP/E-YFP in HEK293 cells deficient in endogenous VRAC. These measurements demonstrated that the time course of FRET changes correlated with the current activation and inactivation when applying hypotonic buffer and consecutively isotonic buffer (new Figure 1F). This replaces the citation of the reviews at this position. So far, we were not able to measure currents from LRRC8A/A homomeric channels – perhaps due to the extremely low conductance (Gaitán-Peñas et al., 2016; Deneka et al., 2018).

Voltage-clamp experiments show that VRAC currents inactivate in response to sustained applied stimulus. It is important to discuss the sustained nature of the FRET signal if it is related to channel activation. Is it thought that there are distinct gates for each modality of channel activation?

As also seen in our simultaneous FRET/current measurements, the FRET signal (reduced FRET, correlating to active VRAC) follows the currents (which we measure at -80mV). Sustained VRAC activity in hypotonicity is not unusual both in whole-cell voltage-clamp mode (e.g. reviewed by Pederson et al., 2016, and shown in their Figure 1A) and cells not subjected to whole-cell patch clamp (as seen in numerous experiments investigating VRAC-mediated osmolyte efflux). Time-dependent inactivation by non-physiological depolarized voltages is typical for VRAC currents (with the inactivation kinetics and voltage-dependence varying with LRRC8 subunit composition – Voss et al., 2014, Ullrich et al., 2016). We observed this characteristic inactivation in our voltage-step protocol measurements that we routinely conducted to confirm that the currents we measured were mediated by VRAC (new Figure 1F and new Figure 4C). There are likely distinct gates for activation/inactivation by osmolarity and depolarization-induced inactivation, but a discussion on this is beyond the scope of this study.

Reviewer #2:For editorsThe authors developed a highly interesting assay to monitor activity of LRRC8 mediated VRAC channels using a purely optical FRET based assay. Such an assay could be useful for structure-function studies, as well as for studies of VRAC function in physiological settings. However, a major criticism is that the assay has not been validated by parallel electrophysiological measurements. In case the authors can provide such validation I would be in favor of publication.

We thank the reviewer for the generally positive comments. Following the reviewers’ suggestions, we have now performed a series of electrophysiological measurements, also in parallel to our FRET assay.

For authorsThe manuscript by König et al., reports a method to monitor VRAC/LRRC8 volume sensitive channel activity in a time- and space-resolved manner by measuring intra-complex FRET between C-terminally fused fluorescent proteins. Two major aims are pursued: first, to demonstrate the flexibility of the leucine-rich-repeat domains (LRRDs) and their involvement in the gating of the channel; second, to provide evidence that low intracellular ionic strength is not the physiologically relevant parameter for the opening of VRACs. Rather, the authors propose that activation under hypotonic stimulation is coupled to PKC activity.In addition, the authors use the assay to show that LRRC8 mediated VRAC is not activatable in endomembranes, but only in the plasma membrane.The crucial questions driving the work are laudable and potentially very important for the field. In particular, the method of introducing an intramolecular FRET sensor of subunit distance is highly innovative and potentially very useful. The possibility to monitor LRRC8 activity with a purely optical assay is very intriguing.

We thank the reviewer for these positive comments. Indeed, we now see a major advantage of a purely optical assay – we do not dialyze the cell and lose signaling pathways.

However, several technical controls are necessary to fully validate the assay. Most importantly, it is mandatory that the assay itself, and other results, are corroborated by electrophysiological data.All the experiments should be complemented with electrophysiological recordings. In principle, it would be most convincing if the FRET assay could be performed on the same cell that is patch-clamped to verify that FRET changes reflect current activation. Since this maybe technically too demanding, it is at least necessary to validate the constructs used by electrophysiological recordings, under the same conditions as the FRET assay. Attaching fluorescent proteins to the C-terminus might alter the "volume-sensitivity" by itself. In fact, the addition of fluorescent proteins to the C-terminus resulted in constitutive anion channel activity in Xenopus oocytes (Gaitán-Peñas et al., 2016).

As suggested by the reviewers, we have now performed patch-clamp fluorometry and measured FRET and current in the same cells. We found that (as previously described by Gaitán-Peñas et al., 2016) the tagged constructs yielded residual currents in isotonicity. Nonetheless, the currents were clearly potentiated by hypotonicity (as shown with measurements in oocytes by Gaitán-Peñas et al., 2016). This rendered the measurements indeed technically demanding (with a gigaseal rate below 20% overexpressing cells versus more than 50% success rate in wild-type cells), as also stated by Gaitán-Peñas et al., 2016: “However, we found this to be extremely difficult because of a very low success rate of giga-seal formation. In the few successful recordings, constitutive VRAC-like channel activity could be observed (data not shown).” Nonetheless, by using patch-clamp fluorometry we could convincingly confirm the relationship of the observed FRET changes with the activation of VRAC.

The authors propose a central role for PKC in the gating of the channel and indicate that its pharmacological inhibition prevented hypotonicity-driven VRAC opening. Again, this result should be verified by electrophysiological recordings employing the same constructs, and constructs without tag – a quantitative comparison of the current density with and without application of PMA / Gö6983 / staurosporine is necessary.

We first confirmed that PMA also induced a reduction in cFRET in HEK293 in addition to HeLa cells (new right panel in Figure 4A) and that cFRET changes were specific for VRAC, since an unrelated FRET sensor (CFP-18aa-YFP) did not respond to PMA (new Figure 4—figure supplement 1A). Subsequently, we have performed electrophysiological experiments on cells overexpressing our fluorescently-tagged LRRC8 constructs and on wild-type cells with endogenous VRAC to investigate the role of DAG (and PMA)-activated kinases.

Remarkably, and at first in apparent contradiction to our results from the FRET measurements, we found that PMA did not activate VRAC in clamped cells (new Figure 4B). Neither the current, nor –surprisingly- the cFRET were affected by PMA application. However, when we incubated the cells with PMA before establishing the whole-cell configuration, we detected activation of fluorescently-tagged VRAC and of endogenous VRAC, as shown in new Figure 4C. Therefore, we could validate our cFRET data by electrophysiological methods and could show for the first time that whole-cell patching of cells affects signaling in the VRAC activation pathway, possibly by cell dialyzes. Electrophysiological measurements further verified the inhibiting effect of DOG (inhibitor of the DAG kinase) on VRAC inactivation after hypotonicity for both fluorescently tagged LRRC8s and for endogenous VRAC (new Figure 5C), corroborating the role of DAG signaling in VRAC activation.

Our attempts to confirm the effect of Gö6983 on cFRET by electrophysiological measurements led to paradoxical results. In cells held in the whole-cell patch-clamp configuration, we observed a tremendous increase in cFRET by a factor of four, which we simply cannot explain. In parallel, Gö6983 did not inhibit VRAC currents, in fact falsifying the proposed role of PKC in VRAC activation. Indeed, we would have missed this point without performing the electrophysiological control experiments suggested by the three reviewers. We have changed all respective statements in the manuscript accordingly. Instead, application of Gö6983 reduced the inactivation of currents when we changed buffers from hypotonic to isotonic. We show these data in new Figure 4—figure supplement 2, but avoid drawing strong conclusions.

Having ruled out PKC as a core mediator of VRAC activation by diacylglycerol (or PMA), we tested PKD, which can also be recruited to the membrane and activated by DAG (and PMA). In contrast to Gö6983, the PKD inhibitor CRT 0066101 reduced hypotonicity-induced VRAC currents both during clamping and when applied before membrane breakthrough (new Figure 5A,B).

Reviewer #3:Volume-regulated anion channels (VRACs) play a central role in the modulation of cell volume in animals, by enabling the flow of chloride ions and larger osmolytes out of the cell upon activation by hypotonic conditions, and therefore counteracting cell-swelling. These channels are heterohexamers composed of the obligatory subunit LRRC8A and additional LRRC8-B-E subunits. The mechanism of activation of VRAC in not understood, but it has been proposed to be related to conformational changes at its C-terminus, and has been shown to be triggered by reduced ionic strength in a reconstituted system. In the present manuscript, Benjamin Koenig and collaborators construct a FRET-based VRAC-activity sensor by fusing the C-termini of different subunits with either of two fluorescence proteins capable of FRET. The authors show changes in FRET in response to hypotonic media that are reversible upon return to isotonic conditions, with kinetics and pharmacology that is consistent with the signal originating from VRAC activation. They also show that hypotonicity-driven FRET changes occur only for channels in the plasma membrane, whereas those located at intracellular compartments (ER and golgi) don't exhibit changes in FRET. They demonstrate that the requirement for activity of localization in the plasma membrane does not involve differences in cholesterol concentration or the integrity of the actin cytoskeleton. In contrast, the authors find that stimulation of PKC activity leads to prominent FRET changes similar to those produced by hypertonic media, whereas inhibition of PKC prevents FRET changes in response to hypotonic media. The authors have developed a set of clever tools to follow VRAC activity without requiring electrophysiological measurements, and provide relevant information on the cell-physiological conditions that enable VRAC activity and on the role of the C-terminus in channel activation. However, I think the manuscript would be much stronger if the authors provided more direct evidence that the FRET changes are indeed connected with channel activation. In addition, they should address some apparent inconsistencies within their observations and also relative to published data, and provide additional information on some experiments.

We thank the reviewer for the generally positive comments. We have addressed the reviewer’s concerns as described below.

I have the following specific concerns:1) All conclusions in the manuscript depend on the assumption that the observed FRET changes are reflecting channel activation. However, the evidence supporting this is not direct, but rather depends on correlative observations of the effects of non-specific treatments that could also affect other processes in the cell. It is possible that the FRET changes reflect a conformational change that is independent or at least partially independent of activation. This issue is particularly important given that the role of the C-terminus for activation has not been well established. I think the manuscript would be significantly stronger if the authors could provide more direct evidence for this, by performing electrophysiological measurements in combination with FRET measurements, or by using additional specific VRAC inhibitors or performing experiments with loss-of-function VRAC subunits.

As suggested by the reviewers, we have now performed electrophysiological measurements simultaneously with cFRET measurements. The new results demonstrate a clear correlation of cFRET changes with VRAC currents. This shows that the FRET sensor is a useful tool to study VRAC activity and that the C-termini move during VRAC activation. We do not claim a strict role of the C-terminus in VRAC activation.

2) One of the main observations of the paper is that localization in the plasma membrane is required for the observed FRET changes and channel activation. However, there is a significant amount of intracellularly located LRRC8 subunits even without treatment with BFA (see LRRC8E-RFP in Figure 2C, for example), and the hypotonicity-dependent FRET signals shown in Figure 1 appear to originate from the entire cell (see Figure 1D). The authors should address this apparent contradiction, and provide detailed information on the way they performed their analysis to measure signals mostly originating from the plasma membrane. The observation that LRRC8E appears to be more heavily retained in the ER that LRRC8A (see Figure 2C) seems to counter the observation that the 8A/8E ratio does not correlate with the observed changes in FRET, except if the majority of the measured FRET signal originates from the plasma membrane. The authors should discuss these issues.

We thank the reviewer for bringing up this issue. Indeed, a portion of either LRRC8 subunit is frequently stuck in the secretory pathway, as we observed previously for exogenously expressed LRRC8 combinations (Voss et al., 2014). The amount of signal that is not from the plasma membrane showed a high variability, with some cells showing much LRRC8A and some cells LRRC8E intracellularly. Usually, there is a tendency towards LRRC8E being more stuck in the endomembrane system, but importantly, we did not observe a correlation with the expression levels. As VRACs can adopt varying subunit stoichiometries, this suggests that the amount of correctly paired (and trafficked) subunits is unaffected by the expression levels as well. Consistently, we did not observe a correlation between absolute acceptor-normalized cFRET values and LRRC8E expression levels (not shown). Indeed, the YFP signal originating from a part of the cell with little CFP (LRRC8A) will not contribute to the real FRET signal of correctly formed VRAC channels (see lack of co-localization in Figure 2C), but it would indeed lower the acceptor-normalized cFRET (see equation in Material and methods section). However, as we did not observe a correlation between expression levels and intracellular localization, there is no correlation between the expression ratio of 8A/8E and the decrease in cFRET (Figure 1—figure supplement 2B). We agree with the reviewer that this is an important issue that deserves clarification in the manuscript, and have therefore added a corresponding statement (subsection “Inter-subunit FRET shows C-terminal movement during VRAC activation”).

3) The authors should discuss the apparent contradiction of their data with the observations in a reconstituted system where decreased ionic strength is sufficient to activate VRAC channels.

Indeed, Syeda et al., (2016) observed activation of LRRC8 complexes reconstituted in membrane bilayers by reduced ionic strength. So far, we have no explanation for this apparent contradiction, except for the environment of the VRACs, i.e. artificial membrane bilayer system versus cellular context. In the cellular context, however, there are several studies providing evidence for VRAC activation without alteration of the ionic strength. Furthermore, alterations of ionic strength required to activate VRACs are usually not reached in experiments using the whole-cell configuration to study VRAC. This has recently been reviewed in detail by Strange et al., (2019). Following the reviewer’s suggestion, we have now expanded our discussion on this, also adding further citations of important studies on the regulatory role of ionic strength on VRAC activation (Discussion section).

4) From the images in Figure 3B, it is impossible for me to determine which channels are localized at the ER vs golgi, even though the authors used a marker for golgi expression. The authors should use ER, golgi and plasma membrane markers, and provide a semi-quantitative analysis to demonstrate the location of the different channel populations, and provide a more detailed description of how they selected their ROIs for performing group analysis of channels in each of the cellular compartments.

We have now added images (new Figure 3—figure supplement 1) showing the co-localization of LRRC8A-CFP-FM2 with the ER marker ER-YFP, the Golgi marker GalNAc-RFP, and with CD4-YFP as plasma membrane marker after release from the aggregation block. We provide a semi-quantitative analysis for a representative cell in which we follow the LRRC8A-CFP-FM2 construct through the secretory pathway (Figure 3—video 1; quantifications shown for each time frame), for which we measure the CFP fluorescence intensity in the Golgi area and a ROI for plasma membrane (depicted in the video). We have now added a more detailed description of the ROI selection in subsection “Reverse aggregation system”.

5) In Figure 1—figure supplement 1, the authors should show the net FRET rather than the normalized values, to allow readers to see how much variability there was in FRET between the controls and the VRACs.

Net FRET values with our fluorescently-labelled VRAC subunits varied tremendously, with measured FRET efficiencies ranging between 10% and 80%. This is likely due to the flexible subunit stoichiometry of hexameric VRAC complexes, at least when expressed exogenously LRRC8 (e.g. Gaitán-Peñas et al., 2016). Accordingly, cFRET (i.e. the corrected FRET value normalized to the acceptor level) varied within the cell population, making a normalization of the values necessary. The relative reduction in cFRET during VRAC activation was independent from the net FRET of a given cell, just as it neither correlated with the subunit expression ratio (see point 2 and Figure 1—figure supplement 2B). Therefore, we decided not to show net FRET values, since this would not add valuable information. However, we now state the variability of FRET as the reason for the normalization (subsection “Acceptor photo-bleaching, sensitized-emission and ratiometric FRET assays”). The net FRET values for the FRET control constructs (CFP-18aa-YFP and GluA2-6Y-10C), on the other hand showed little variability between cells, but we also present them as normalized values for clarity.

6) The authors should show whether the treatments with LatB or MbCD affected net FRET levels, rather than just the normalized values.

Net FRET values vary extremely between cells explained (see our answer to point 5). The mean of this highly variable value was not altered by the treatment with LatB or MbCD: we observed mean (non-normalized) cFRET of 52 ± 15% (untreated cells in isotonic buffer), 45 ± 17% (LatB) and 47 ± 12% (MbCD). We opt not to present these data in the manuscript, since we think it will not provide further useful information.

7) The Abstract, Title and the overall text seem to imply that PKC activation is connected with the physiological activation of VRACs upon exposure to hypotonic solutions. However, the presented data does not really establish a link between VRAC stimulation by PKC and changes in extracellular osmolarity. It might very well be that PKC-stimulated VRACs are indeed activated by ionic strength, and that this sensitivity is obscured by PKC inhibition. The authors should provide a more lengthy discussion on how PKC might be affecting channel activity, even if based only on speculation.

This aspect of the manuscript has changed substantially following new patch-clamp measurements and the realization that PKC is not as central as we suspected. Instead, we show a much stronger connection to DAG-PKD signaling, because we can activate, inhibit or sustain VRAC currents in the absence of, or in opposition to, hypotonicity.